# Peptide derived from SLAMF1 prevents TLR4-mediated inflammation in vitro and in vivo

Kaja Elisabeth Nilsen[1], Boyao Zhang[2], Astrid Skjesol[1], Liv Ryan[1], Hilde Vagle[1], Maren Helene Bøe[1], Pontus Orning[2], Hera Kim[1], Siril Skaret Bakke[1], Kirusika Elamurugan[1], Ingvild Bergdal Mestvedt[1], Jørgen Stenvik[1,3], Harald Husebye[1], Egil Lien[1,2], Terje Espevik[1,3,*], Maria Yurchenko[1,3,*]

Inflammation plays a crucial role in the development and progression of many diseases, and is often caused by dysregulation of signalling from pattern recognition receptors, such as TLRs. Inhibition of key protein–protein interactions is an attractive target for treating inflammation. Recently, we demonstrated that the signalling lymphocyte activation molecule family 1 (SLAMF1) positively regulates signalling downstream of TLR4 and identified the interaction interface between SLAMF1 and the TLR4 adaptor protein TRIF-related adapter molecule (TRAM). Based on these findings, we developed a SLAMF1-derived peptide, P7, which is linked to a cell-penetrating peptide for intracellular delivery. We found that P7 peptide inhibits the expression and secretion of IFN$\beta$ and pro-inflammatory cytokines (TNF, IL-1$\beta$, IL-6) induced by TLR4, and prevents death in mice subjected to LPS shock. The mechanism of action of P7 peptide is based on interference with several intracellular protein–protein interactions, including TRAM–SLAMF1, TRAM–Rab11FIP2, and TIRAP–MyD88 interactions. Overall, P7 peptide has a unique mode of action and demonstrates high efficacy in inhibiting TLR4-mediated signalling in vitro and in vivo.

## Introduction

TLRs are one of the five major groups of pattern recognition receptors (PRRs) in innate immune cells. Each TLR recognizes distinct microbial components that are shared among groups of microbial species, so-called pathogen-associated molecular patterns. As immune sensors, the TLRs provide broad protection against a variety of potential pathogens. They also recognize damage-associated molecular patterns, which are molecules of endogenous origin that may function as alarm signals for tissue damage or abnormalities (1). Binding of pathogen-associated molecular patterns (or damage-associated molecular patterns) to the N-terminal ligand recognition domain of PRRs induces dimerization and conformational changes to the TLRs, allowing binding of adaptor proteins to the intracellular TIR domains. These adaptors connect the TLRs to a network of intracellular signaling pathways that eventually lead to the activation of transcription factors and changes in gene expression (2).

TLR4 is the major sensor of Gram-negative bacteria (3, 4, 5). LPS or endotoxin is a component of the outer membrane of most Gram-negative bacteria, and is recognized by TLR4 via serum-localized LPS-binding protein and CD14, which transfer LPS to a complex of TLR4 and myeloid differentiation factor 2 (MD-2) at the cell surface (6). LPS binding to TLR4/MD-2 triggers dimerization of the ecto-domain and structural changes in TLR4 which lead to TIR–TIR dimer formation. In this agonistic TLR4 conformation, the TIR–TIR dimers bind to the sorting adaptor protein TIRAP which recruits the signaling adaptor MyD88. The detailed mechanism of TLR4 activation is not fully resolved on the atomic level, but a recent simulation study suggests a dynamic and plastic behaviour of TLR4, which depend on the lipid environment (lipid rafts), and formation of two possible types of functional TIR–TIR dimers (symmetric and asymmetric) (7). After initiation of MyD88/TIRAP-dependent signaling from the plasma membrane, which leads to the expression and secretion of pro-inflammatory cytokines like TNF and IL-6, TLR4 dimers undergo endocytosis (2, 8). Here, the sorting adaptor TRAM binds TLR4 dimers and recruits the signaling adaptor TRIF, which, via TRAF3 and TBK1 (and IKK$\varepsilon$), induces the activation of the transcription factor IRF3, which mediates induction of type I IFN expression (2). Among the TLRs, TLR4 is unique in its ability to activate both MyD88- and TRIF-dependent signaling pathways (8).

Under normal circumstances, the immune responses induced by TLRs confer an effective limitation of infections and appropriate induction of the adaptive responses. However, dysregulation or

[1]Centre of Molecular Inflammation Research, Department of Clinical and Molecular Medicine, Norwegian University of Science and Technology, Trondheim, Norway [2]Program in Innate Immunity, Division of Infectious Diseases and Immunology, Department of Medicine, University of Massachusetts Chan Medical School, Worcester, MA, USA [3]Department of Infectious Diseases, Clinic of Medicine, St. Olavs Hospital HF, Trondheim University Hospital, Trondheim, Norway

Correspondence: mariia.yurchenko@ntnu.no
*Terje Espevik and Maria Yurchenko contributed equally to this work

inappropriate activation of TLR-mediated responses can be detrimental to the host, contributing to the development of autoimmune diseases and sepsis, to name a few (9). Sepsis is a life-threatening clinical syndrome involving an exaggerated systemic inflammatory response to pathogens with excessive production of pro-inflammatory mediators (10), which could develop into septic shock. This severe physiological state is characterized by circulatory, cellular, and metabolic abnormalities, which could result in multiple organ failure and cardiovascular collapse (10, 11). Recently, researchers (12) provided evidence for the contribution of the TLR4-TRAM-TRIF signaling pathway to the late-phase sepsis pathology, more specifically to kidney injury. They observed a global shutdown of protein synthesis, importantly affecting vital metabolic pathways, which was linked to TLR4-mediated TRAM-TRIF–induced genes (12). Overall, targeting TLR4 signaling, particularly TLR4-TRAM-TRIF–dependent signaling, could be beneficial for the treatment of sepsis patients.

In recent years, the development of decoy peptides based on TIR domains of TLR signaling molecules has shown promise in reducing inflammatory responses in the early stages of testing (13). A peptide named TIP1, derived from the TIR domain of TIRAP, inhibited cytokine secretion downstream of several TLRs, and rescued mice from LPS-induced sepsis and kidney/liver damage (14). The same group also developed a second peptide based on the TIR domain of TIRAP, TIP3 that was inhibitory towards TLR3 and TLR4 (15). Another peptide derived from the TIR domain of TLR2 inhibited inflammatory responses downstream of several TLRs and improved survival in an influenza mouse model (16). This research group also demonstrated that several peptides derived from the TIR domain of TRAM reduced LPS-induced inflammatory responses and protected mice from septic shock (17). A potential challenge with TIR domain-derived peptides could be a lack of specificity because of the high homology of TIR domains in TLRs, adaptor proteins, and IL-1R. The TIR domain-derived peptides mentioned above were extensively tested in murine cells. Thus, another aspect to consider is potential species differences in efficacy of such peptides. Nonetheless, the approach of using peptides that have intracellular targets and are delivered to the cells together with cell-penetrating peptides (CPPs) has garnered a lot of interest and could have great therapeutic potential for many diseases, including inflammation, cancer, diabetes, etc. (18).

We have recently identified a new regulatory protein in the TLR4-mediated TRAM-TRIF–dependent signaling pathway in human immune cells, namely the signaling lymphocytic activation molecule family 1 (SLAMF1), a type I transmembrane glycoprotein of the CD2-like family of proteins. We have demonstrated that SLAMF1 is required for the trafficking of TRAM from the endocytic recycling compartment to *Escherichia coli*-containing phagosomes in human macrophages, where TRAM binds dimerized TLR4 and recruits TRIF, resulting in the induction of *IFNβ* mRNA expression. We have identified the interaction domains of SLAMF1 and TRAM and shown that the SLAMF1-TRAM protein–protein interaction (PPI) is essential for the regulation of TLR4-TRIF-TRAM–dependent signaling and IFNβ production in human innate immune cells (19). Therefore, we decided to exploit this knowledge and target SLAMF1-TRAM PPI using a SLAMF1-derived peptides based on the TRAM-interacting sequence of SLAMF1. Considering the potential detrimental role of

TLR4-TRAM-TRIF signaling in sepsis (12, 20, 21) and excessive TLR4 signaling in other inflammatory diseases (reviewed in references 1, 22, 23, 24, 25, 26), this peptide could be a new therapeutic agent with favorable properties compared with previously developed TIR-domain–derived peptides or TIR domain-targeting small molecules (14, 15, 16, 17, 27). In the current study, we developed a SLAMF1-derived peptide named P7, which interferes with TLR4-mediated signaling in human cells, whole blood, and prevents animal death in a murine endotoxemia in vivo model system. We demonstrate that P7 inhibits TLR4 signaling by targeting several crucial PPIs in the signaling pathway.

## Results

### Designing and testing SLAMF1-derived peptides in THP-1 cells

The TRAM-interacting sequence in SLAMF1 (19) was used to design several peptide candidates with the aim of inhibiting the interaction between SLAMF1 and TRAM, and subsequently, the TLR4-TRAM-TRIF–dependent signaling pathway.

To prove the concept and limit the size of the peptide, ECFP-tagged peptides were overexpressed in HEK 293T cells. Subsequently, cell lysates were used in immunoprecipitation assays (IPs) to test the peptides' ability to inhibit SLAMF1-TRAM co-precipitation in vitro. A panel of ECFP-based constructs were generated (Fig 1A) where the ECFP-tagged P1, P2, P3, P6, and P10 contain a part of the human SLAMF1 sequence (NCBI database, EAW52706, amino acids 318–335), and the P7 and P11 contain a P333T substitution corresponding to the minor SNP allele in the human *SLAMF1* gene (rs3796504, minor allele frequency 0.04, Ensembl).

HEK 293T lysates with overexpressed ECFP constructs were normalized for the comparable level of ECFP or ECFP–peptide expression and mixed with lysates containing overexpressed TRAM[FLAG] and SLAMF1 proteins, followed by anti-FLAG IPs. The IPs demonstrated that ECFP-tagged peptides interfered with the SLAMF1–TRAM interaction (Fig 1A). P6 and P7 peptides were the smallest peptides that efficiently blocked SLAMF1–TRAM interaction, with no apparent effect of the P333T substitution in these experimental settings (Fig 1A). Still, when comparing the minor SLAMF1[FLAG] P333T variant with the major (WT) variant, the minor variant appeared considerably more effective in co-precipitating TRAM in the overexpression system (Fig 1B). Thus, even though the initial experiment with the peptides did not reveal a major difference, peptides with the P333T substitution might be more efficient for blocking TRAM–SLAMF1 interaction and could have a competitive advantage over endogenous SLAMF1 in most human subjects. Therefore, we have proceeded with testing peptides with P333T substitution, with P7 as lead candidate because of its shortest size combined with efficacy.

Peptides that obstruct intracellular PPIs possess tremendous therapeutic potential (28). However, delivering these peptides intracellularly has challenges. Most synthetic peptides cannot penetrate the plasma membrane without a CPP sequence. CPPs are short peptides (usually 5–30 amino acids) that can cross the cell membrane and when combined with a functional peptide

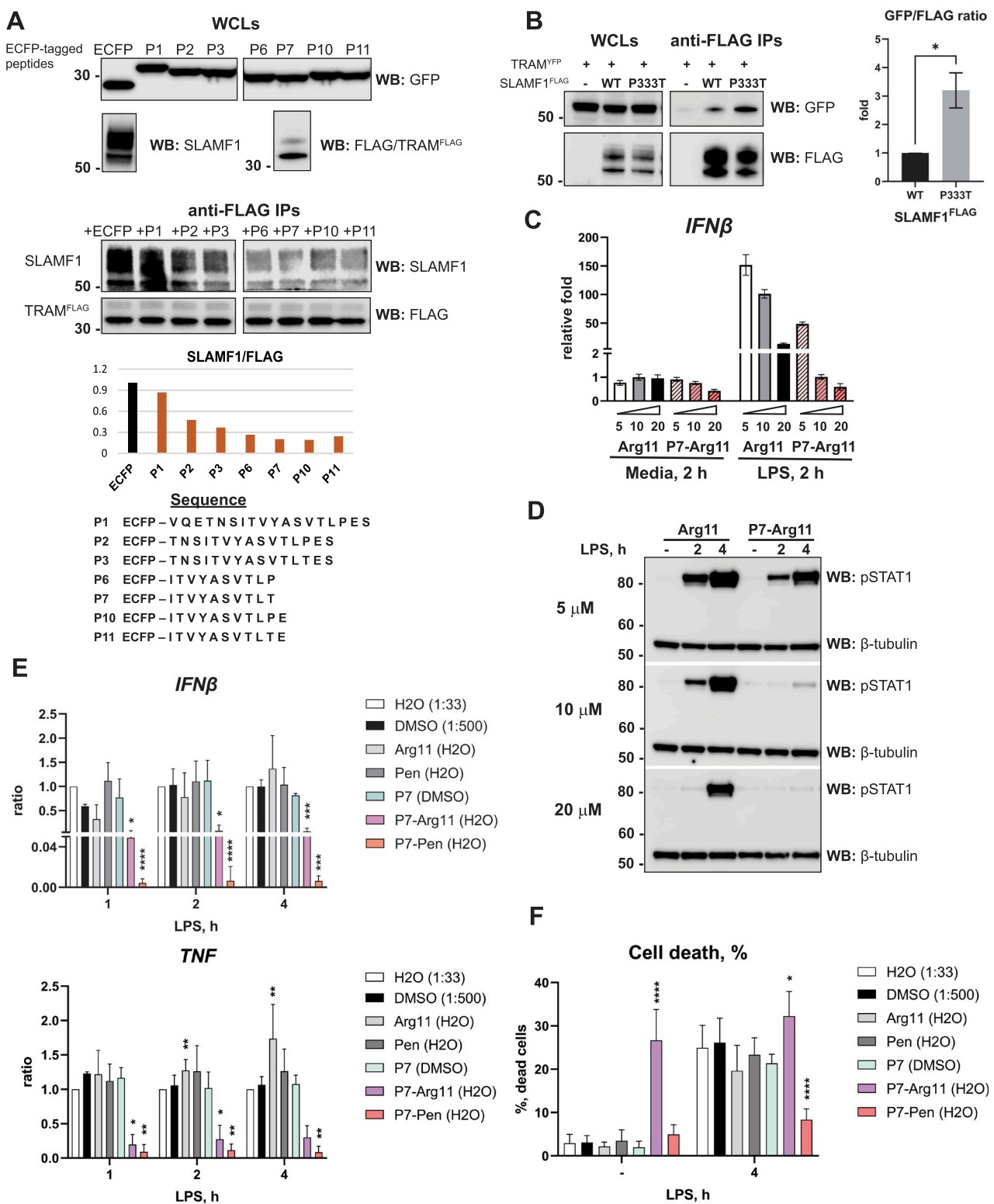

**Figure 1. SLAMF1-derived peptide P7 is blocking TRAM-SLAMF1 co-precipitation and efficiently inhibits TLR4-mediated signaling in THP-1 cells when used in a complex with cell-penetrating peptide Arg11 or Pen.**

**(A)** HEK 293T cells were transfected by constructs coding for ECFP or ECFP-tagged peptides (sequences listed in the table), or SLAMF1 or TRAM$^{FLAG}$ proteins. After 48 h, cell lysates were prepared, and lysates containing ECFP- or ECFP-tagged peptides were normalized for the comparable expression levels, which were controlled by anti GFP WB (top panel). Whole-cell lysates from TRAM$^{FLAG}$ or SLAMF1 overexpressing cells were also analyzed by WB. Lysates with ECFP and ECFP peptides were mixed with equal amounts of lysates with overexpressed TRAM$^{FLAG}$ and SLAMF1, and incubated with anti-FLAG beads for 2 h, followed by WB analysis of co-precipitated proteins. Samples

sequence, CPPs act as vehicles or vectors for intracellular delivery and targeting of peptides (29). We selected two commonly used CPPs—11 poly-arginine (Arg11) and Penetratin (Pen) as CPP candidates for peptides' intracellular delivery. The synthesis of peptides was commissioned from specialized providers, adhering to stringent criteria for high purity (>90–95%), and necessitating the conversion of the toxic TFA salt to acetate salt after synthesis. This conversion is essential for the utilization of the peptide with living cells. We conducted pilot experiments to determine the effect of synthetic Arg11-linked P7 at different concentrations on TLR4-mediated *IFNβ* expression to select the concentration range for future assays. Pretreatment of differentiated THP-1 cells with P7-Arg11 efficiently reduced LPS-mediated *IFNβ* mRNA expression (Fig 1C), and LPS-induced phosphorylation of STAT1 at 10–20 $\mu$M (Fig 1D). Phosphorylation of STAT1 Tyr701 is induced by IFNβ-interferon-$\alpha$/$\beta$ receptor (IFNAR) signaling (30, 31), and can thus serve as a marker of IFNβ secretion in the THP-1 cell culture. Because 20 $\mu$M of the Arg11 control peptide alone reduced both IFNβ transcription and secretion (Fig 1C and D), possibly because of toxicity or nonspecific effects, we proceeded with 15 $\mu$M peptides and investigated cell viability in parallel with cytokine production. Inhibition of LPS-mediated STAT1 phosphorylation by P7 in THP-1 cells directly correlated with *IFNβ* mRNA expression (Fig 1C and D), thus, we further proceeded with qRT-PCR analysis for evaluation of the peptide's efficacy.

Furthermore, we compared the efficacy of the CPPs Arg11 and Pen as vectors for P7. Differentiated THP-1 cells were pretreated with peptides for 30 min, followed by LPS stimulation for 1, 2, and 4 h, and cytokine induction was determined by qRT–PCR (Fig 1E), with normalization against water control. A representative image showing typical gene expression fold change (normalized to unstimulated control water sample) for one of the experiments included to Fig 1E is demonstrated in Fig S1A.

Overall, both P7-Arg11 and P7-Pen, but not Pen and Arg11 or P7 without CPP, significantly reduced LPS-mediated *IFNβ* and *TNF* mRNA expression (Figs 1E and S1A), with P7-Pen being the most potent inhibitor. The potential toxicity of the peptides was evaluated by LDH release assay. LPS stimulation itself induces cell death in PMA-differentiated THP-1 cells, resulting in 20–30% of cell death in 4 h of stimulation (Fig 1F, water control). P7-Arg11 exhibited high toxicity even in unstimulated cells, whereas P7-Pen alone had no toxic effect and significantly inhibited LPS-mediated cell death (Fig 1F). Therefore, Pen was chosen as CPP for P7 peptide.

Furthermore, we designed and tested a panel of Pen-linked peptide variants (Table S1) with single amino acid substitutions or size modifications (P10 and P11 peptides with one additional amino acid). We included to this panel a control peptide (C3) with four simultaneous amino acid substitutions when compared with P7 peptide. To pinpoint the amino acids that are essential for the interaction between the peptide and target proteins, we performed tests utilizing peptides where each individual amino acid in the P7 sequence was replaced with alanine—a small, nonpolar amino acid with a short side group. In addition, we explored various substitutions involving nonpolar amino acids, specifically valine and alanine (at positions 3 and 5, respectively), by replacing them with leucine—a nonpolar amino acid characterized by a larger side group. THP-1 cells were pretreated by 12.5 $\mu$M peptides or solvents ($H_2O$ or DMSO) and stimulated by LPS. LDH release and the levels of CXCL10 regulated by IFNβ via IFNAR in THP-1 cells (19), TNF, and IL-1β cytokines were measured in cell culture supernatants after 4 h of LPS stimulation. Ratio for the LDH or cytokine release for each peptide to the respective solvent was calculated and presented as a heatmap in Fig S1B.

Most of the tested alanine substitutions led to a reduction in the inhibitory activity of the SLAMF1-derived peptide during the screening process (Fig S1B). Similarly, substitutions of nonpolar amino acids with polar amino acids, such as serine and threonine, also resulted in a notable decline in the peptide's inhibitory effectiveness (Fig S1B). Meanwhile, positions 4 and 10 within the P7 peptide exhibit greater flexibility and could potentially be used to introduce diverse chemical modifications, aiming to enhance the peptide's stability in biological fluids. Overall, P7-Pen was the most effective inhibitor of LPS-mediated CXCL10, TNF, IL-1β secretion, and LDH release among the whole tested panel.

### The SLAMF1-derived peptide P7 specifically inhibits TLR4-mediated signaling in primary human monocytes

To investigate the specificity of the P7 peptide, we tested its ability to alter cytokine mRNA expression and secretion triggered by TLR2, -4, and STING signaling in human primary monocytes or THP-1 cells. P7 significantly reduced TLR4-mediated *IFNβ*, *TNF*, *IL-6*, and *IL-1β* mRNA expression, and TNF, IL-6, MIP-1$\alpha$/CCL3, IL-10, and CXCL10 secretion in LPS-treated human monocytes (Fig 2A and B). Because of the absence of inflammasome activation signal that is required for IL-1β cleavage and release by monocytes (32), IL-1β secretion was not induced by any of the treatments, and only some decrease

shown for anti-GFP, anti-SLAMF1, and anti-TRAM WBs were loaded and transferred to the same membranes, with several bands excised on the presented images. Quantification of SLAMF1 protein to TRAM[FLAG] ratio in IPs is shown on graph. One of three independent experiments. **(B)** Anti-FLAG IPs for WT or P333T mutant SLAMF1[FLAG] with YFP-tagged TRAM protein. Input (Whole-cell lysates) comprised of 7.5% from the sample used for IP. Correlation of GFP signal intensity to FLAG signal intensity is shown on the graph. Data presented as mean ± SD for three independent experiments, significance evaluated by $t$ test with Welch's correction (*$P$ < 0.05). **(C)** Quantification by qRT-PCR of *IFNβ* mRNA expression in THP-1 cells pretreated for 30 min with variable concentrations of control Arg11 or P7-Arg11 peptides (5, 10 or 20 $\mu$M), followed by stimulation with LPS (100 ng/ml). Results presented as mean ± SD for three biological replicates (one of three experiments). **(C, D)** Western blot analysis of p-STAT1 expression in THP-1 cells pretreated with peptides and stimulated with LPS as in (C). $\beta$-tubulin WB used for loading control. **(E)** Quantification of *IFNβ* and *TNF* mRNA expression by qRT-PCR in THP-1 cells pretreated by peptide solvents (water, $H_2O$ or DMSO) or peptides (15 $\mu$M) for 30 min and stimulated with LPS for indicated time points. Data for each time point are normalized to water ($H_2O$) and presented as ratio between mRNA expression values of cells treated with peptide to values for cells treated with water. Data presented as mean relative fold change + SD (data from 6–11 independent experiments). **(E, F)** Cell death evaluated by LDH release assay in supernatants of cells used for qRT-PCR analysis in (E). Data presented as mean for percentage of dead cells + SD. **(E, F)** Statistical testing was done by mixed effect model on log-transformed data (*$P$ < 0.05, **$P$ < 0.01, ***$P$ < 0.001, ****$P$ < 0.0001).
Source data are available for this figure.

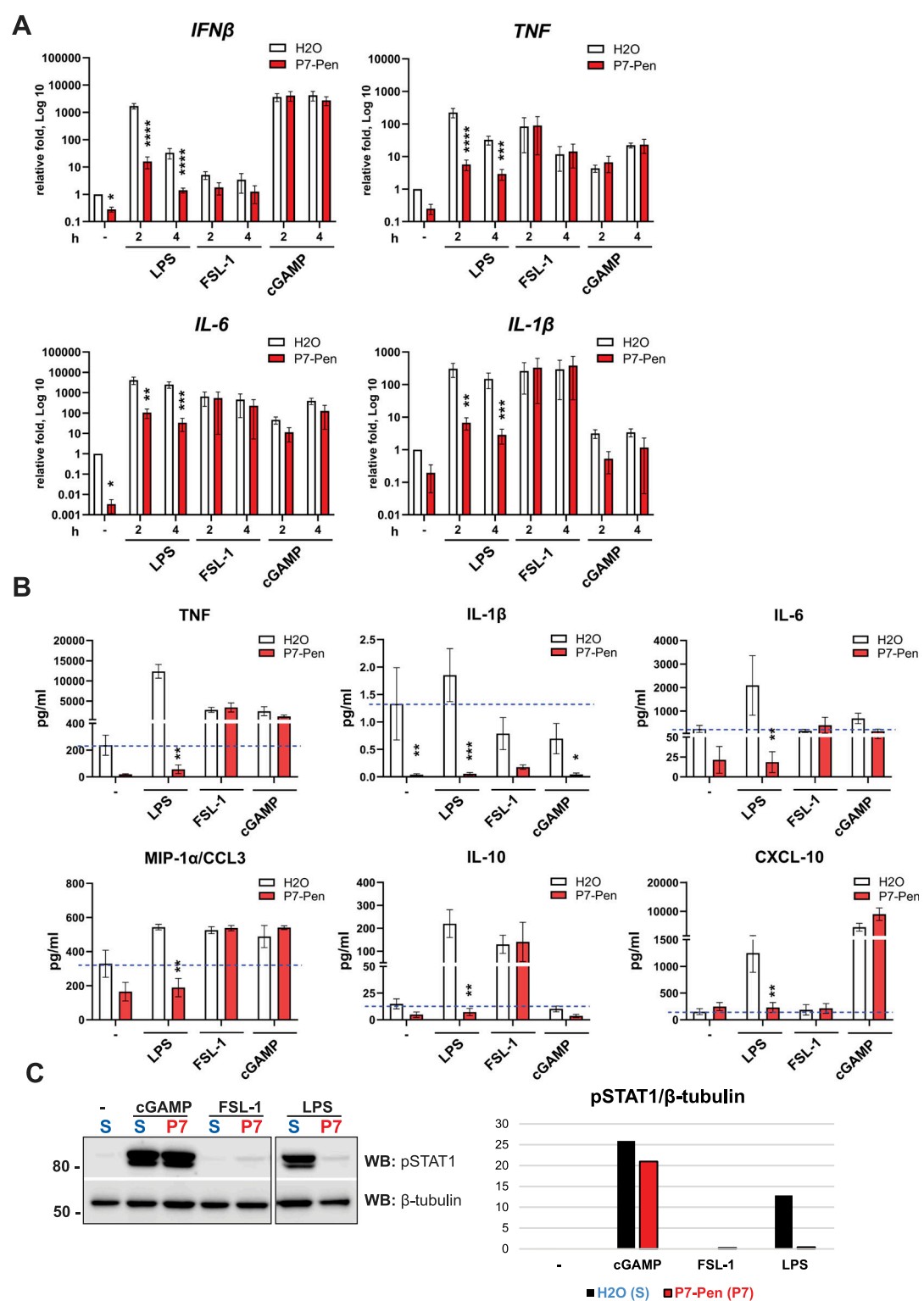

**Figure 2. In primary human monocytes, P7 inhibits TLR4-mediated pro-inflammatory cytokines and IFNβ expression and secretion, whereas having no effect on cytokines downstream STING and TLR2 signaling.**

**(A, B, C)** Primary human monocytes isolated from PBMCs from healthy donors were pretreated with water solvent (H₂O) or 15 μM P7-Pen peptide for 30 min, followed by stimulation with LPS (100 ng/ml), FSL-1 (100 ng/ml) or cGAMP (2′,3′-cGAMP) (20 μg/ml) for 2 or 4 h, followed by collection of supernatants (for cytokines secretion analysis), and cell lysis (for simultaneous isolation of total RNA for qRT–PCR and protein for WB analysis). **(A)** Quantification of *IFNβ*, *TNF*, *IL-6*, and *IL-1β* mRNA expression by qRT-PCR in primary human monocytes. Data presented as fold change when compared with unstimulated sample pretreated by water (H₂O), mean ± SEM (n = 6–11), in log scale. Statistical testing was done by two-way ANOVA or mixed effects model on log-transformed data (*$P < 0.05$, **$P < 0.01$, ***$P < 0.001$, ****$P < 0.0001$). **(B)** Cytokine secretion

of the IL-1β background levels was observed for the cells pretreated with P7 (Fig 2B). The phosphorylation of STAT1 was greatly reduced by P7 in cells stimulated by LPS, which was tested as a readout for IFNβ/IFNAR signaling (Fig 2C). The decrease of pSTAT1 level in P7-treated cells after LPS stimulation reflected the robust reduction of TLR4-mediated *IFNβ* mRNA expression (Fig 2A) and was in line with the inhibition of TLR4-mediated CXCL10 secretion (Fig 2B), which is positively regulated by pSTAT1. P7 had no significant effect on TLR2- or STING-mediated cytokine mRNA expression or secretion, and neither on STING-mediated STAT1 phosphorylation (Fig 2A–C). An LDH release assay indicated that all treatments and stimulation conditions had not affected cell viability (Fig S2).

Altogether, our results demonstrate that the SLAMF1-derived peptide P7 efficiently attenuates TLR4 but not TLR2 or STING signaling. Hence, P7 likely has additional targets in the MyD88-dependent signaling pathway.

### P7 inhibits LPS and *E. coli*-mediated cytokine secretion in human whole-blood model

Peptides lacking specific modifications typically have a short lifespan in human blood because of their nonspecific binding and neutralization by plasma proteins or rapid renal filtration (33). To determine whether the P7 peptide would remain effective in inhibiting TLR4-mediated cytokine release in human blood, we used an established human whole-blood model with lepirudin as an anticoagulant. Lepirudin does not interfere with PRR signaling and preserves the active complement system, thus maintaining experimental conditions that closely resemble physiological conditions (34). Peripheral anticoagulated blood from healthy volunteers were pretreated with solvent or peptides at several concentrations before the addition of LPS (100 ng/ml) or *E. coli* particles (1 × 10^6/ml) for 5 h. At the end of stimulation, plasma samples were separated by centrifugation for further ELISA or BioPlex analysis of cytokines. Like the observations made for human primary monocytes (Fig 2), P7 significantly reduced LPS-induced IFNβ, IL-1β, TNF, IL-6 secretion at all tested concentrations (10, 20 and 40 μM), and MIP-1α/CCL3, and IL-8 secretion at 20 and 40 μM concentrations (Fig 3A). Similarly, P7-Pen exhibited a significant reduction in *E. coli*-mediated secretion of IFNβ and IL-1β across all tested concentrations. However, its inhibitory efficacy was less pronounced in the case of *E. coli*-mediated TNF and IL-6 secretion, with significant reductions observed only at the highest tested concentrations of P7-Pen (Fig 3B). Notably, there was minimal alteration in the levels of MIP-1α/CCL3 and IL-8 secretion. *E. coli* bioparticles could trigger recognition by various PRRs found within immune cells in whole blood, including TLR4, TLR1/TLR2, TLR2/TLR6, intracellular nucleic acid sensors, and the complement system (35, 36, 37). The relatively diminished effectiveness of P7-Pen in inhibiting *E. coli*-mediated cytokine secretion, as compared with LPS-

mediated signaling, could be attributed to its selective inhibition of TLR4-mediated signaling while not affecting complement-, TLR2-or other PRR-mediated pathways. Our results clearly demonstrate that the P7 peptide retains its ability to effectively reduce TLR4-mediated IFNβ and pro-inflammatory cytokine release by immune cells of whole blood, whether stimulated with the TLR4 pure ligand (LPS) or heat-killed *E. coli* (bioparticles).

### P7 inhibits TRAM interaction with SLAMF1 and FIP2, but not with TRIF or TLR4

The effect of P7 is most prominent on inhibition of TLR4-induced *IFNβ* mRNA expression and secretion (Figs 1E, 2A, and 3). Our previous studies demonstrated that the TLR4 signaling pathway leading to IFNβ secretion is positively regulated by SLAMF1 receptor (19). The SLAMF1-derived P7 peptide was designed to inhibit TRAM–SLAMF1 interaction, which was confirmed in the initial IP screens of ECFP-tagged peptide candidates (Fig 1A). We further proceeded with co-precipitation assays using synthetic peptides in HEK 293T cells overexpressing TLR4, TRAM, TRIF, Rab11 family interacting protein 2 (FIP2) or SLAMF1. In these assays, we tested the effect of C3-Pen (control) or P7-Pen peptides on TRAM PPIs with the known binding partners TLR4, TRIF, SLAMF1, and FIP2. As expected, the co-precipitation of SLAMF1 with TRAM was strongly reduced in the presence of P7-Pen compared with the control peptide (Fig 4A). At the same time, P7-Pen did not inhibit the co-precipitation of TRAM with either TLR4 or TRIF (Fig 4B and C), like what was previously shown for SLAMF1 protein itself (19). Briefly, whereas SLAMF1 silencing significantly attenuates TRAM recruitment to bacterial phagosomes, SLAMF1 overexpression and interaction with TRAM does not interfere with TLR4–TRAM–TRIF complex formation in HEK 293T cells (19). We have also demonstrated earlier that both SLAMF1 and TRAM interact with FIP2, and the FIP2 interaction site in TRAM protein partially overlaps with the SLAMF1 interaction site (19, 38). Therefore, it was important to test whether the TRAM–FIP2 interaction could be affected by P7. Indeed, anti-FLAG IPs from cells overexpressing FIP2^EGFP and TRAM^FLAG demonstrated that P7-Pen significantly reduced the amount of FIP2 co-precipitated with TRAM (Fig 4D). These data indicate that P7 interferes with TRAM–FIP2 interaction, which is instrumental for driving bacterial phagocytosis and intracellular trafficking of TRAM to bacterial phagosomes (38).

### P7 inhibits TRAM and FIP2 recruitment to *E. coli* phagosomes and *E. coli* phagocytosis

Next, we hypothesized that P7, which disrupts TRAM interaction with SLAMF1 and FIP2, would inhibit the recruitment and accumulation of TRAM around *E. coli* phagosomes. To test this hypothesis, THP-1 cells overexpressing TRAM^CHERRY protein or primary human monocytes

---

levels addressed by BioPlex assays for TNF, IL-1β, IL-6, MIP-1α, IL-10, and CXCL-10, and presented as mean ± SEM (n = 5–8). Basal level in unstimulated water pretreated cells is shown as a dash blue line on the graph. Statistical significance evaluated using Mann–Whitney test. **(C)** WB for pSTAT1 (Tyr701) levels after stimulation for 4 h by different PRR ligands for the cells pretreated by water solvent control (S) or P7-Pen peptide (P7) and normalized to endogenous control β-tubulin (quantification on the graph). Representative image for one out of five donor samples.
Source data are available for this figure.

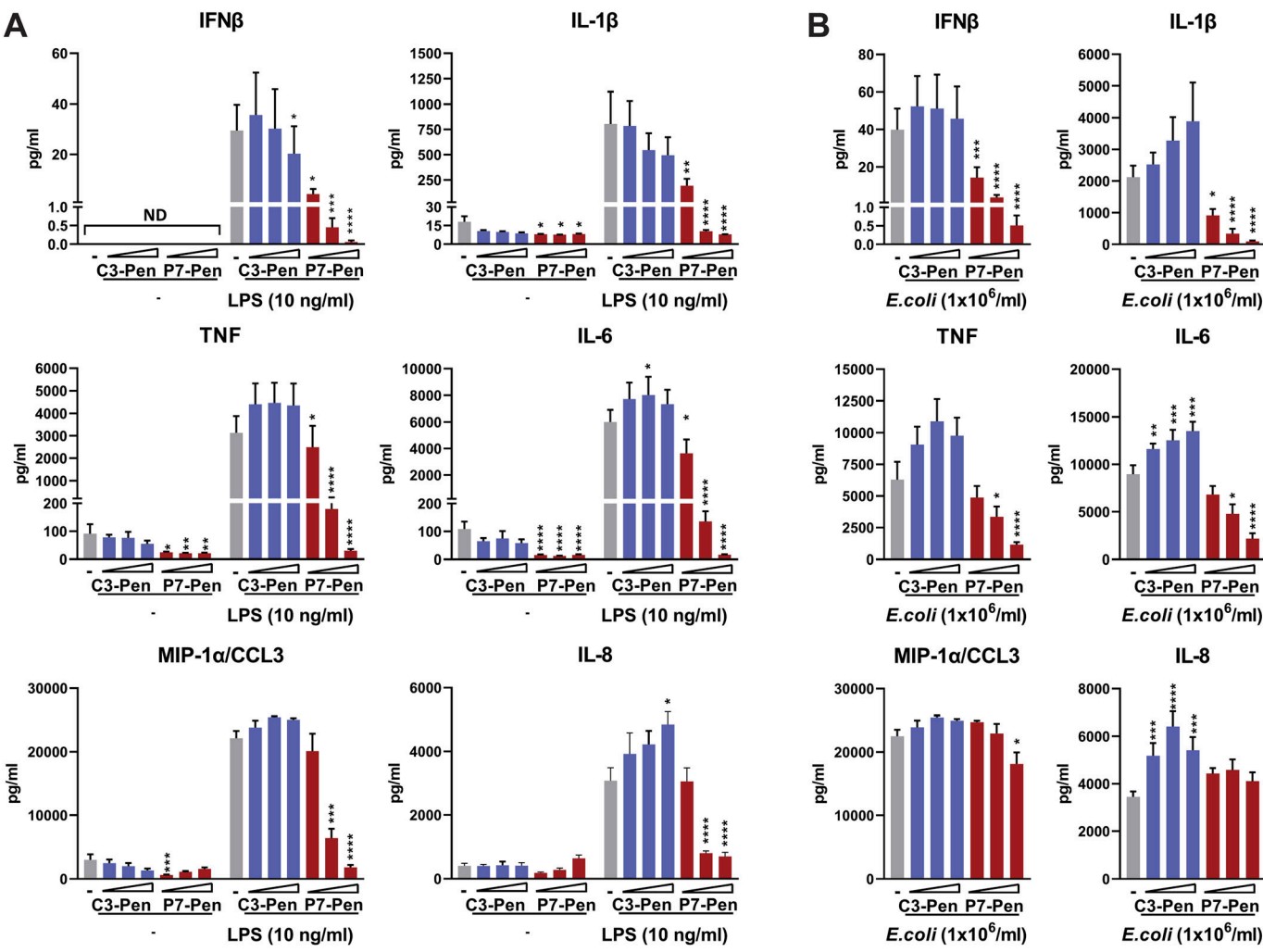

**Figure 3.  P7-Pen, but not the control peptide C3-Pen, significantly inhibits LPS and *E.coli*-mediated IFNβ and pro-inflammatory cytokine release in a human whole-blood model in a concentration-dependent manner.**

Whole-blood assay for blood samples of a healthy donor with lepirudin as an anticoagulation reagent. **(A, B)** Blood samples were pretreated with water solvent, or control (C3-Pen) and P7-Pen peptides at 10, 20, and 40 $\mu M$ concentrations for 30 min, followed by addition of LPS (100 ng/ml) (A) or *E.coli* particles ($10^6$/ml) (B) for 5 h before the collection of plasma samples. Plasma samples were probed for IFNβ secretion by ELISA, and for IL-1β, TNF, IL-6, MIP1-α, and IL-8 secretion using BioPlex assays. Data presented as mean ± SEM, statistical significance evaluated using Wilcoxon matched-pairs signed-rank test, significance levels (*$P < 0.05$, **$P < 0.01$, ***$P < 0.001$, ****$P < 0.0001$, nonsignificant if not shown otherwise). ND, not detected.

were treated with either P7 or a control peptide before being stimulated with Alexa Fluor 488 *E. coli* particles. After fixation and staining of cells with anti-TRAM antibodies, the cellular localization of TRAM was investigated by confocal microscopy. We had to stain TRAM with anti-TRAM antibodies not only in primary cells, by also in THP-1 TRAM^CHERRY cells (Fig 5B), because CHERRY fluorescent signal was almost undetectable after fixation of cells (whereas being strong in the live cells (38)). Accumulation of TRAM around *E. coli* particles was assessed by measuring the mean voxel intensity (MVI) of TRAM staining around the bacterial particles (Fig 5A–C). As can be seen in Fig 5A and C, P7 significantly reduced the recruitment of TRAM to *E. coli* particles in both THP-1 cells and human primary monocytes, which supported our hypothesis.

We have previously shown that the interaction between TRAM and FIP2 is critical for TLR4-mediated phagocytosis (38).

P7 disrupted TRAM interaction with FIP2 (Fig 4D), therefore, we questioned whether the recruitment of FIP2 to bacterial particles would be inhibited by P7. To investigate this, we performed confocal microscopy of primary human monocytes pretreated with P7 or a control peptide and incubated cells with *E. coli* AF488 particles for 30 min, fixed cells and stained with anti-FIP2 antibodies (Fig 5D and E). Quantification of FIP2 MVI around phagocytosed *E. coli* particles demonstrated that P7 significantly reduced the recruitment of FIP2 to bacterial phagosomes (Fig 5D).

Furthermore, we addressed whether *E. coli* uptake was also altered by P7 treatment. THP-1 WT cells or human monocytes were stimulated with *E. coli* particles in the presence of P7 or a control peptide (Pen), or cytochalasin D (CytoD) for the indicated time (Fig 5F and G). Quantification of particles inside the cells revealed that P7 reduced bacterial uptake as efficiently as CytoD, whereas the control peptide had no such

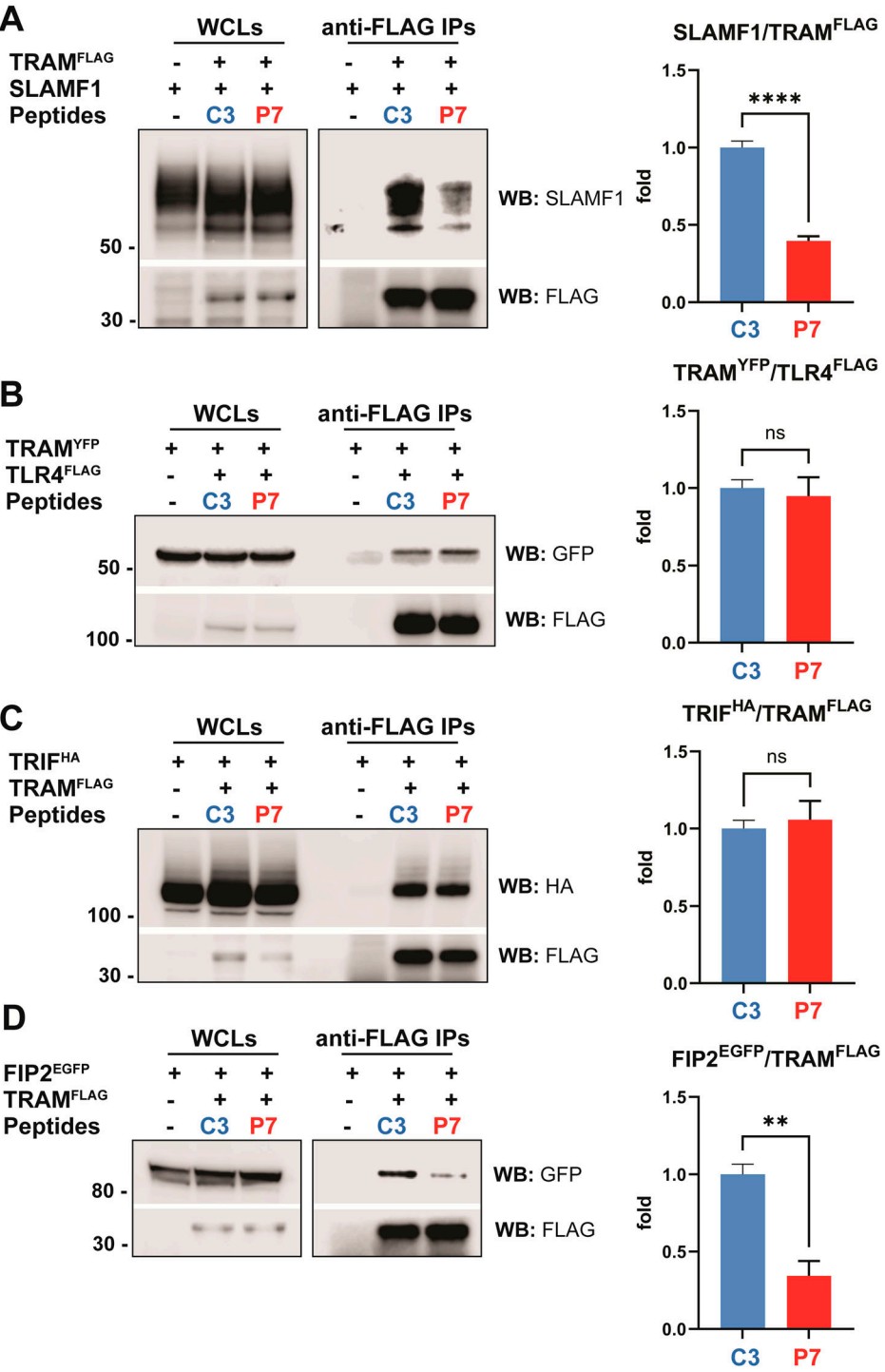

Figure 4.   P7-Pen peptide inhibits SLAMF1-TRAM and Rab11 FIP2-TRAM co-precipitation, whereas not affecting TLR4-TRAM and TRIF-TRAM co-precipitation in HEK 293T cells.
(A, B, C, D) HEK 293T cells were co-transfected by SLAMF1 and TRAM$^{FLAG}$ (A), TRAM$^{YFP}$ and TLR4$^{FLAG}$ (B), TRIF$^{HA}$ and TRAM$^{FLAG}$ (C) or FIP2$^{EGFP}$ and TRAM$^{FLAG}$ (D) for 48 h, followed by treatment of cells by 30 $\mu$M peptides C3-Pen (control, C3) or P7-Pen (P7) for 1 h, lysis of cells and anti-FLAG IPs for 3 h. Whole-cell lysates loaded for the input control, where input represents 7.5% from the total sample used for IP. (A, B, C, D) Ratio between co-precipitated proteins to FLAG-tagged proteins was quantified for three to four independent experiments and presented on graphs to the right from the respective WB (A, B, C, D). Significance evaluated by t test with Welch's correction, significance levels (**P < 0.01, ****P < 0.0001, ns, nonsignificant).
Source data are available for this figure.

effect (Fig 5F and G). These results were further supported by flow cytometry-based uptake assays using *E. coli* pHrodo red particles, which confirmed a significant reduction in the percentage of *E. coli* pHrodo-positive cells (Fig 5H) and a significant decrease of pHrodo median fluorescent intensity (Fig 5I) by P7 when compared with the controls. These results indicate that P7 not only interferes with TRAM–SLAMF1 and TRAM–FIP2 PPIs and the subsequent recruitment of TRAM to the TLR4 on *E. coli* phagosomes, but also inhibits bacterial uptake.

## Complement-driven *E. coli* phagocytosis is not inhibited by P7

Opsonization of pathogens by the complement system greatly enhances phagocytosis (39, 40, 41). The experiments presented above were conducted using bacterial particles that were not opsonized before the experiment. Hence, the impact of the complement system on phagocytosis was minimal. To establish if P7 could inhibit phagocytosis in more physiological settings, where the

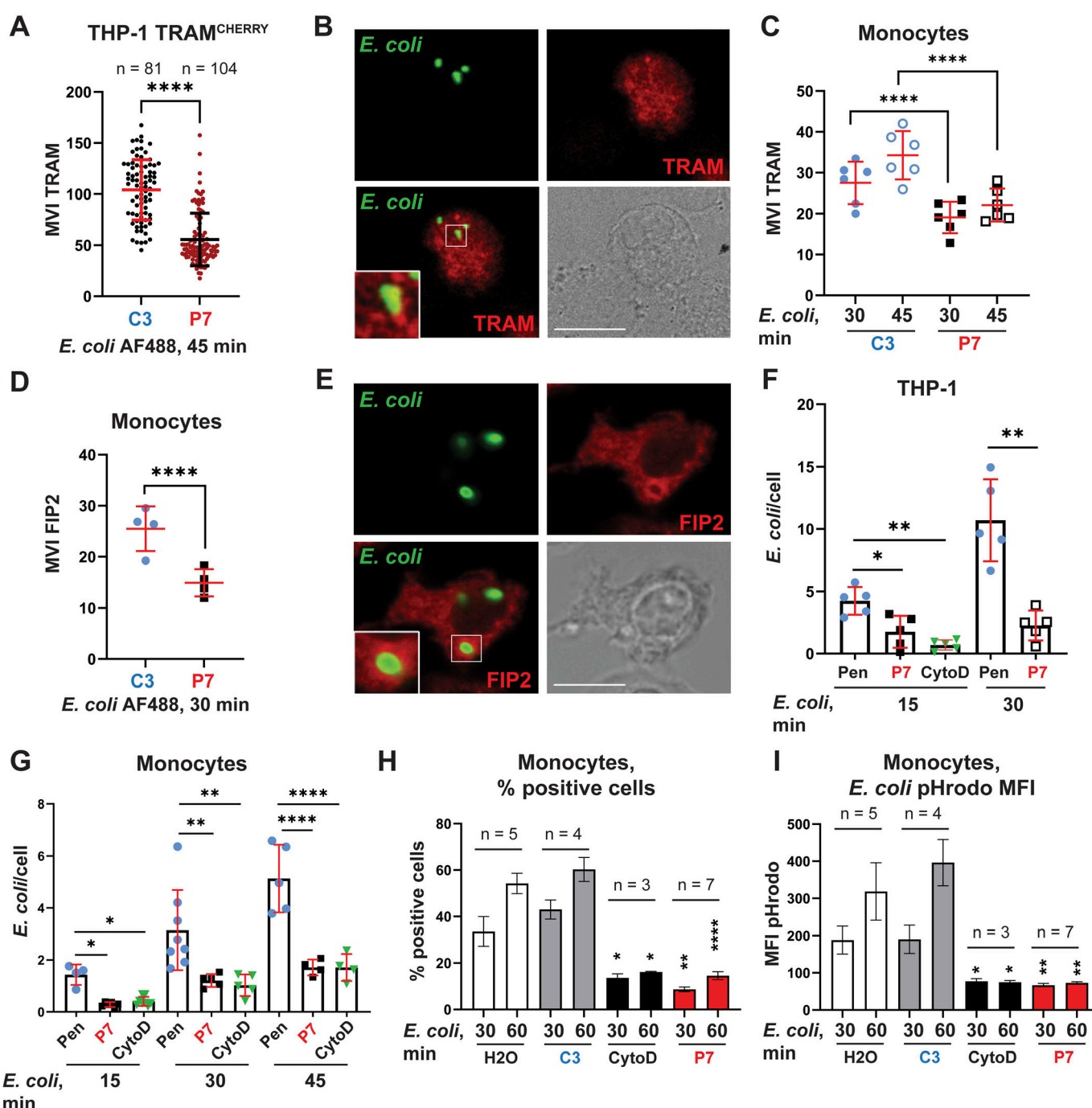

**Figure 5.** **SLAMF1-derived peptide P7 inhibits TRAM and FIP2 recruitment to *E. coli* phagosomes and *E. coli* phagocytosis in THP-1 cells and primary human monocytes.** **(A, B, C, D, F, G)** THP-1 cells overexpressing TRAM THP-1 TRAM [CHERRY] (A, B), THP-1 WT (F) or primary human monocytes (C, D, G) were pretreated by 15 μM control (C3-Pen) or P7-Pen peptides for 30 min and incubated with *E. coli* AF488 particles for the indicated time points. **(A, B, C, D, E, F, G)** Cells were fixed, stained for TRAM (A, B, C) or FIP2 (D, E), followed by confocal microscopy imaging (A, B, C, D, E, F, G). **(A)** TRAM mean voxel intensities for individual phagosomes quantified from xyz images from three independent experiments. Data presented as mean ± SD, statistical significance evaluated by Mann–Whitney test. **(B)** Representative image for TRAM (red) and *E. coli* AF488 (green) staining in THP-1 TRAM [CHERRY] cells. Scale bar 10 μm. **(C)** TRAM MI on *E. coli* phagosomes, each dot represent mean value for individual donor (n = 6), quantified from five xyz images (20–30 cells per image) for each donor. Data presented as mean ± SD, statistical significance evaluated by two-way ANOVA. **(D)** FIP2 mean voxel intensities on *E. coli* phagosomes, each dot represents the mean value for individual donor (n = 4), quantified from five xyz images for each donor. Data presented as mean ± SD, statistical significance evaluated by two-way ANOVA. **(E)** Representative image for FIP2 (red) and *E. coli* AF488 (green) staining in monocytes, scale bar 10 μm. **(F, G)** Quantification of average *E. coli* particles per cell for THP-1 wt cells (F) or primary human monocytes (G), representative experiments (n = 3). Cells were pretreated by 15 μM control peptide Pen or SLAMF1-derived peptide P7-Pen or CytoD (3 μM). CytoD used as a control for inhibition of bacterial uptake. Individual dots represent mean value from quantification of particles for one xyz image (20–30 cells per image). Statistical significance evaluated by one-way ANOVA. **(H)** Quantification of phagocytosis based on flow cytometry for primary human monocytes pretreated by a solvent (water, H₂O), or 15 μM peptides, or 3 μM CytoD for 30 min and incubated with *E. coli* pHrodo particles for indicated time points. **(H, I)** Percentage of *E. coli* pHrodo red-positive cells shown on (H) and median pHrodo fluorescence intensity on (I). Data presented

complement system is active, we proceeded with assays using opsonized bacterial particles. First, to directly compare the effect of opsonization conditions on phagocytosis, we set up bacterial uptake assays using human primary monocytes, where *E. coli* pHrodo particles were opsonized in normal human serum (with active complement system), or serum containing the complement inhibitor compstatin (42, 43) or heat-inactivated (h.i.) serum (complement activity inactivated). Flow cytometry analysis showed that both heat inactivation of serum and the addition of compstatin markedly reduced the uptake of bacterial particles compared with normal serum (Fig S3A). Next, we assessed the effect of P7 on phagocytosis of bacterial particles opsonized by normal or h.i. serum. P7 treatment only slightly reduced particle uptake when particles were opsonized by normal serum (active complement), whereas strongly inhibited phagocytosis of particles opsonized with h.i. serum or serum with compstatin (Fig S3B–D). These data indicate that the inhibitory effect of P7 on phagocytosis is largely overcome by opsonization in physiological conditions. Interestingly, different opsonization conditions did not affect much *IFNβ* mRNA expression by the cells from parallel wells after 1 h of stimulation by bacteria (Fig S3E), and P7 significantly reduced *IFNβ* mRNA expression despite opsonization conditions (Fig S3E). Finally, TRAM recruitment to *E. coli* particles opsonized by either normal or h.i. serum was significantly reduced by P7 (Fig S3F), which goes in line with the inhibitory effect of P7 on *IFNβ* mRNA expression for all opsonization conditions (Fig S3E).

Taken together, these results provide several interesting insights. First, by interfering with TRAM–FIP2 interaction, P7 acts at the early stage of TLR4-mediated phagosomal signaling, leading to reduced complement-independent phagocytosis and inhibition of trafficking of TRAM and FIP2 to *E. coli* phagosomes. Second, inhibition of TRAM–SLAMF1 interaction by P7 contributes to the reduction of TRAM recruitment to *E. coli* phagosomes and consequent inhibition of TLR4–TRAM–TRIF-dependent signaling. Therefore, when opsonization conditions minimize the effect of P7 on phagocytosis (Fig S3B), *IFNβ* mRNA expression is still inhibited by P7, as seen in assays using primary human monocytes (Fig S3E) and the whole blood model (Fig 3B).

### P7 interferes with MyD88-dependent TLR signaling by blocking TIRAP–MyD88 interaction

The TLR4-mediated pro-inflammatory response is initiated from the cell surface membrane and is MyD88-dependent (8) (Fig 6A). Induction of TLR4-mediated pro-inflammatory cytokines does not require the recruitment of TRAM adaptor protein (8). However, the P7 peptide significantly reduced TLR4-mediated pro-inflammatory cytokine secretion in THP-1 cells, monocytes, and whole blood assays (Figs 1–3). These data suggest that P7 may target PPIs within the MyD88-dependent pathway in addition to the TRAM-dependent pathway.

After binding ligand and dimerization, TLR4 attracts MyD88 via the TIRAP adaptor protein, forming the Myddosome complex with IRAK4 and IRAK1 (Fig 6A). This is followed by phosphorylation and ubiquitination of IRAK1 and assembly of the signaling complex with TAB1, 2, and TAK1. This signaling complex initiates NF-κB nuclear translocation and activation of mitogen-activated kinases (MAPK) p38 MAPK and JNK1/2, which are required for AP-1 transcription factor activation and nuclear translocation. Overall, this leads to the transcription of pro-inflammatory cytokine genes (Fig 6A).

To determine whether P7 could co-precipitate with Myddosome proteins, we performed pull downs (PDs) from lysates of human monocytes (unstimulated or LPS-stimulated) using biotinylated P7-Pen or control peptide (Pen). Along with the P7 target protein TRAM, several proteins of the Myddosome complex co-precipitated with P7, including the adaptor protein TIRAP, IRAK1, and IRAK4 kinases, and to a much lower extent, MyD88 (Fig 6B). Pull down of these factors were independent of LPS-stimulation, whereas posttranslationally modified IRAK1 (polyubiquitin and phosphorylations) was only present after LPS stimulation.

To reveal which Myddosome protein is directly targeted by P7 and how rapidly this occurs during LPS-stimulation, we treated monocytes with P7-Pen before LPS-stimulation. P7 completely inhibited LPS-induced posttranslational modifications of IRAK1 and downstream phosphorylation of p38 MAPK (Fig 6C and D). We also performed immunoprecipitation of endogenous IRAK4 and IRAK1 from the lysates of LPS-stimulated primary human monocytes and showed that P7 strongly inhibits both IRAK1 and IRAK4 recruitment to MyD88, and hence the Myddosome assembly (Fig 6E and F).

To further clarify the target protein, we investigated whether P7 could inhibit signaling downstream of the IL-1 receptor (IL-1R), because both TLR4 and IL-1R initiate the Myddosome assembly with IRAKs. However, whereas TLR4 recruits MyD88 via the sorting adapter TIRAP, IL-1R recruits MyD88 directly (44). We stimulated HEK 293T IL-1R–expressing cells with human recombinant IL-1β and addressed posttranslational modifications of downstream factors. P7-Pen did not affect IL-1R signaling as revealed at the level of p38 MAPK phosphorylation or polyubiquitination and phosphorylation of IRAK1 (Fig S4A). This indicates that P7 does not interfere directly with the assembly of the Myddosome, for example, by blocking MyD88–IRAK interaction. A possible target of P7 could instead be TIRAP, which is not required for IL-1R-signalling but is involved in proximal TLR4 signaling (45, 46, 47). On the other hand, TLR2 also signals via the TIRAP–MyD88 pathway (48), but P7 did not interfere with TLR2-mediated cytokine production in primary human monocytes (Fig 2). Moreover, P7 neither affected the phosphorylation of TBK1 nor p38 MAPK kinases in monocytes after stimulation with the TLR2 ligand FSL-1, whereas it strongly inhibited LPS-mediated TBK1 and p38 MAPK phosphorylation (Fig S4B). These results were somewhat surprising if to suggest that P7 is blocking TIRAP. However, previous studies have shown that the requirement for TIRAP may vary between different TLRs and could depend on the type and concentration of the ligand (45, 49). In HEK 293T cells, TIRAP overexpression was strictly required for the co-precipitation of MyD88 with TLR4FLAG, whereas MyD88 efficiently co-precipitated with TLR2FLAG also when TIRAP was not overexpressed (Fig S4C and D). It is possible that TLR2 may bind MyD88 directly or that low

as mean ± SEM, statistical significance evaluated by Mann–Whitney test in pairwise comparison with control treatment (solvent, H$_2$O) for the respective time point. Significance levels (*$P < 0.05$, **$P < 0.01$, ***$P < 0.001$, ****$P < 0.0001$).
Source data are available for this figure.

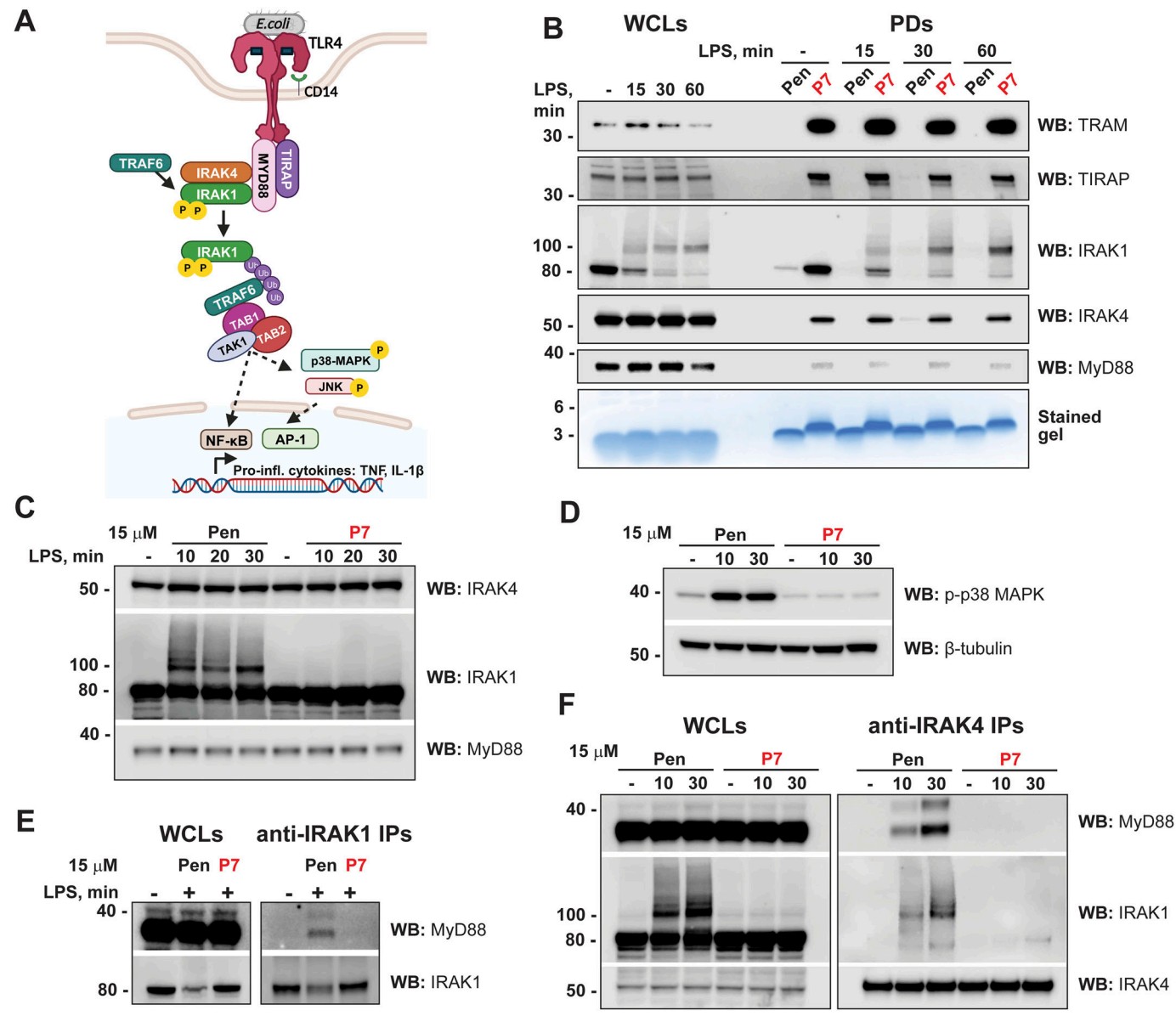

**Figure 6. SLAMF1-derived peptide P7 co-precipitates with endogenous TRAM protein and in addition with key Myddosome-signaling molecules and efficiently abrogates TLR4-mediated IRAK1 and IRAK4 recruitment to MyD88 and IRAK1 posttranslational modifications.**
**(A)** Scheme showing key proteins involved in TLR4–Myddosome assembly upon binding of bacterial LPS ligand. **(B)** WB analysis of proteins that co-precipitated with peptides in pull downs (PDs) from the lysates of unstimulated and LPS-stimulated (100 ng/ml) primary human monocytes. Lower part of the gels used for WB analysis was stained by SimplyBlue SafeStain for peptides' loading control. Input (WCLs) represents 14.5% from the total sample used for PD. Representative experiment is shown from four tested donors. **(C, D)** Western blot analysis of IRAK4 and MyD88 total protein levels, IRAK1 posttranslational modifications (C) and phosphorylation of p38 MAPKs (D) in lysates of monocytes pretreated with peptides and stimulated with 100 ng/ml LPS for indicated time points, representative experiment from a total of five with different donor cells. **(C, D)** Total MyD88 WB or β-tubulin WB were used for loading control (C, D). **(E, F)** Endogenous IRAK1 (E) or IRAK4 (F) were immunoprecipitated by specific Abs covalently fixed on magnetic beads for 4 h from lysates of human monocytes: either untreated cells or cells stimulated by LPS (100 ng/ml) for the indicated time. Input (WCLs) represents 4.6% from the total sample used for IP. Cellular lysates were analyzed for input control, with WBs shown for MyD88, IRAK1, and IRAK4. **(E, F)** Representative experiment from a total of three (E) or four (F) consecutive experiments with different donor cells.
Source data are available for this figure.

levels of endogenous TIRAP in HEK 293T cells is sufficient for TLR2–MyD88 signaling.

We used THP-1 cells overexpressing TLR4[FLAG] and performed TIRAP and FLAG co-IPs and evaluated the effect of P7-Pen on TLR4, TIRAP and MyD88 interaction during LPS stimulation. WB analysis of the precipitates showed that P7 abrogated the co-precipitation of endogenous MyD88 and TIRAP, and the co-precipitation of endogenous MyD88 with TLR4[FLAG], whereas TIRAP co-precipitation with TLR4[FLAG] was not affected by P7 (Fig 7A and B). In line with data from THP-1 cells, endogenous IPs from human monocytes using anti-TIRAP antibodies demonstrated that P7

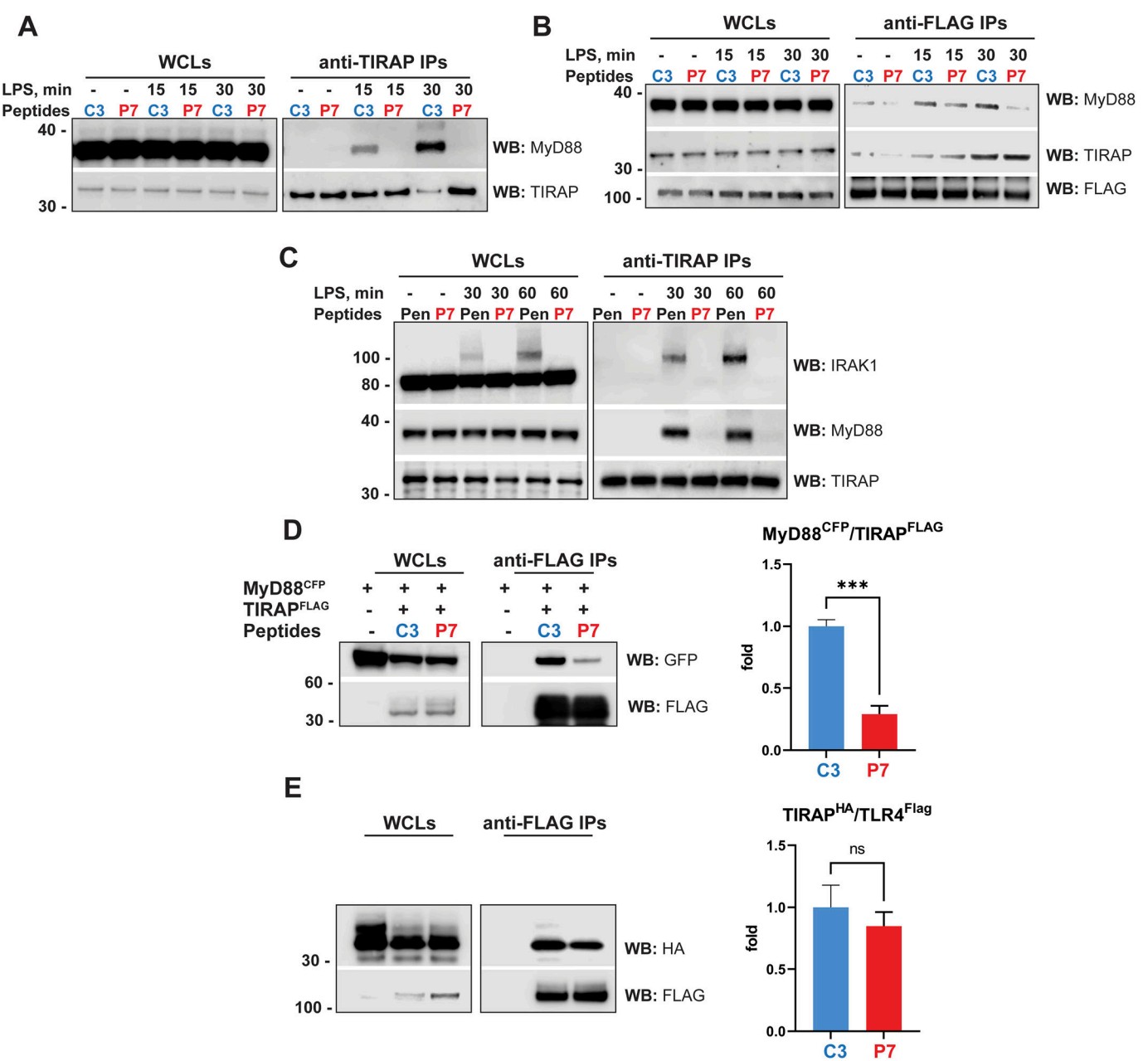

**Figure 7. Interaction of P7 with TIRAP disrupts TIRAP – MyD88 interaction in THP-1 cells, human monocytes, and HEK 293T overexpression model.**
**(A, B)** Endogenous TIRAP (A) or anti -FLAG (B) IPs from lysates of THP-1 TLR4[FLAG] cells. **(C)** Endogenous anti-TIRAP IPs from lysates of primary human monocytes. **(A, B, C)** Cells were pretreated by 15 μM peptides C3-Pen (C3) or P7-Pen (P7) and stimulated by LPS (100 ng/ml) for indicated time, with unstimulated cells used for negative control. **(A, B, C)** Cellular lysates were loaded for input control, with WBs performed for MyD88, TIRAP (A, B, C), FLAG (B), and IRAK1 (C). Input (whole-cell lysates) represents 4.6% from the total sample used for IP. **(A, B, C)** Representative experiments from a total of three for each experimental setting (A, B, C). **(D, E)** Western blot analysis of lysates and anti-FLAG IPs from HEK 293T cells, overexpressing TIRAP[FLAG] and MyD88[CFP] (D) or TLR4[FLAG] and TIRAP[HA] (E), performed in 48 h after transfection and after 1 h of pretreatment of cells by 30 μM peptides. Whole-cell lysates loaded for input control, which represents 14.5% from the total sample used for IP. Ratio between co-precipitated proteins to FLAG-tagged proteins was quantified for three to four independent experiments and presented on graphs to the right from the respective WB. Statistical significance calculated using *t* test with Welch's correction, significance levels: ***P < 0.001, ns, nonsignificant.
Source data are available for this figure.

inhibited MyD88 co-precipitation with TIRAP and the recruitment of IRAK1 to the complex (Fig 7C). Moreover, in the overexpression system, P7-Pen significantly reduced the co-precipitation of MyD88[CFP] with TIRAP[FLAG], whereas the co-precipitation of TIRAP[HA] with TLR4[FLAG] was not much affected (Fig 7D and E). Overall, these results suggest that P7 interrupts the critical interaction of TIRAP and MyD88.

## P7 directly interacts with the N-terminal part of TRAM and TIR domain of TIRAP

P7 co-precipitates with endogenous TRAM and TIRAP proteins (Fig 6B); however, co-precipitation alone cannot be considered a proof of direct interaction between proteins, which may be a part of large protein complex. Therefore, to investigate the direct interaction of the peptide with the target proteins, we proceeded with pull-down assays in a cell-free system using biotinylated peptides Pen, P7-Pen, and P7N4-Pen on avidin beads, and GST-tagged proteins: GST-TRAM (2–100 aa), GST-TIRAP (87–160 aa), GST-TIRAP (30–74 aa), and GST as a negative control.

Because P7N4-Pen (Y4N substitution in P7) was found to be less effective in inhibiting TLR4-mediated cytokine secretion in THP-1 cells (Fig S1B), it served as an additional negative control. The pulled-down proteins were analyzed by gel electrophoresis, visualized by Coomassie G-250 staining of gels and quantified for the GST protein/peptide ratio with BioRad Image Lab software. The results demonstrate a direct interaction between P7 and GST-TRAM (2–100 aa) (Fig 8A and C), and P7 and GST-TIRAP (Fig 8B and C). The P7 interaction could be mapped to the TIR domain of TIRAP (87–160 aa), whereas P7 did not bind GST or GST-TIRAP (30–74 aa) (Fig 8B and C). Control peptides Pen and P7N4-Pen did not interact with GST-TIRAP and GST-TRAM proteins (Fig 8).

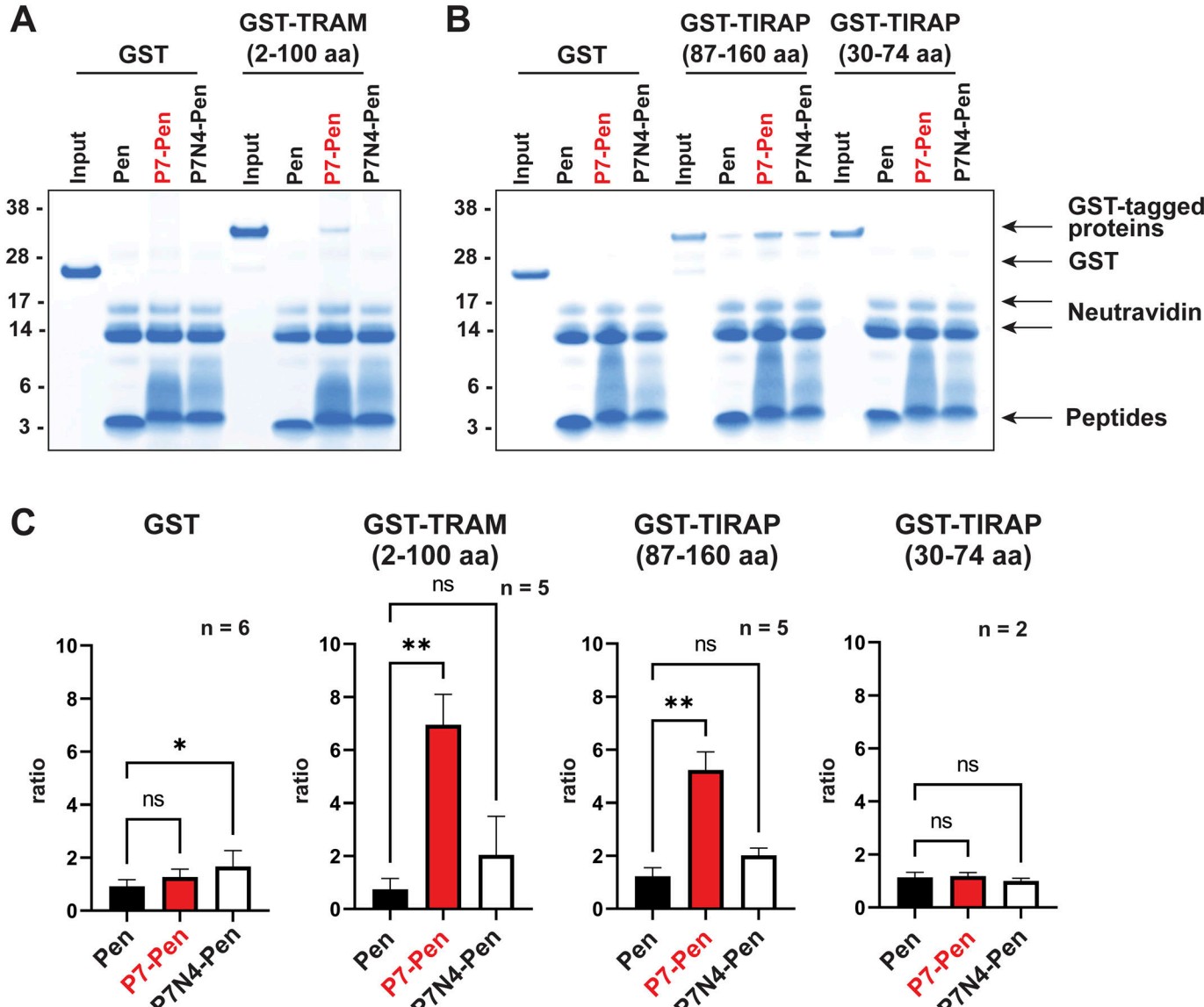

**Figure 8. P7-Pen, but not P7-Pen with Y/N substitution (P7N4-Pen) or Pen CPP directly interacts with GST-TRAM and GST-TIRAP recombinant proteins in cell-free assays.**
**(A, B)** Representative images of stained gels for the pull-down assays (PDs) by biotinylated peptides fixed on NeutrAvidin beads (30 min), to test the interaction of peptides with GST-TRAM (2–100 aa) (A), GST-TIRAP (87–160aa), GST-TIRAP (30–74 aa) (B), and GST (A, B). Input equals the total amount of recombinant protein used for each PD. **(C)** Graphs showing quantification for several PDs, statistical significance evaluated by one-way ANOVA with Dunnett's multiple comparison test, significance levels: *$P < 0.05$, **$P < 0.01$, ns, nonsignificant.
Source data are available for this figure.

### P7 inhibits TLR4-mediated signaling in murine macrophages

Murine disease models are commonly used for in vivo drug testing. We wanted to investigate whether P7, which was designed based on the human SLAMF1 protein sequence, could be used for in vivo studies in mice without requiring species adaptation of the sequence. Previously, we found that only human, but not murine SLAMF1 protein co-precipitated with TRAM. This is because of a sequence difference between human and murine SLAMF1 proteins upstream of the TRAM interacting domain in SLAMF1, which probably changed the orientation of TRAM interacting domain masking it from interaction with murine TRAM (19). Therefore, we questioned whether the SLAMF1-derived peptide P7 might still bind to murine TRAM and function as a TLR4-signaling inhibitor in murine immune cells. Indeed, both human and murine TRAM$^{FLAG}$ proteins co-precipitated with biotinylated P7-Pen when TRAM orthologs were overexpressed in HEK 293T cells. This could be explained by the release of the restrictions of SLAMF1 whole-protein structure in small peptides and by the high homology of SLAMF1-interacting domains in murine and human TRAM (Fig 9A).

To examine if P7 could inhibit TLR4-mediated signaling in murine cells, we used immortalized bone marrow-derived macrophages from B6 mice. Cells were pretreated with P7 or control peptide, and stimulated with LPS, followed by WB analysis of protein lysates and qRT-PCR to assess cytokine gene expression (Fig 9B and C). As shown in Fig 9B, P7 efficiently inhibited the phosphorylation of several key signaling proteins downstream of TLR4, including p38 MAPK, TAK1, IκBα, TBK1. Ubiquitinated murine IRAK1 protein is not detected on WB, therefore, murine IRAK1 posttranslational modification is reflected by the disappearance of the 80-kD band on the anti-IRAK1 WB (Fig 9B). Consistent with these results, the expression of LPS-induced *IFNβ*, *TNF*, and *IL-1β* genes in B6 cells was also significantly inhibited by P7 compared with solvent (water) or control peptide-treated cells (Fig 9C). Overall, we found that P7 efficiently inhibits both TLR4–TIRAP–MyD88 and TLR4–TRAM–TRIF signaling pathways in murine cells as it does in human cells. Therefore, P7 can be directly tested in in vivo murine disease models.

### P7 prevents animal death in murine LPS shock model

LPS shock model is a very relevant proof-of-concept model for evaluation of P7 peptide efficacy for blocking TLR4 signaling in vivo. We have tested P7 in two different experimental settings. In the first round, mice received intraperitoneal (i.p.) injection of LPS or P7-Pen alone, or i.p. P7-Pen or control peptide C3-Pen (2.5 nmol/g) 1 h before LPS injection (20 mg/kg). We found that P7-Pen was not inducing any mortality in mice on its own (100% survival). Mice receiving P7-Pen before LPS injection had significantly improved the survival rate (90%) compared with mice receiving LPS alone (20%) or control peptide with LPS (10%) (Fig 9D). These findings were corroborated by measurements of body temperature, showing that pretreatment with P7-Pen significantly reduced the drop of body temperature mediated by LPS injection (Fig 9E).

In the second round, we compared the pretreatment and post-treatment protocols, where P7-Pen was injected i.p. either 30 min before (2.5 or 5 nmol/g of peptide, pretreatment groups) or 30 min

after LPS injection (5 nmol/g, posttreatment groups) (Fig S5). The study was performed in a different animal facility, and despite using the same LPS as in the first round, this dose of LPS appeared to be much less lethal for the animals and only 36% of mice died in the pretreatment control group and 20% in the post-treatment control group (water/vehicle groups) (Fig S5A and C), whereas none of the mice died in P7 pre and posttreatment groups. P7 had similar effect on the LPS-mediated body temperature drop, showing that mice that received P7-Pen before LPS had a lower drop in body temperature after LPS injection and recovered to normal temperature faster (Fig S5A). As to the body weight (BW), mice treated with P7-Pen in pre or post-treatment groups recovered to normal BW faster than respective water-treated control groups (Fig S5A and C). Plasma samples of all animals (pre and posttreatment groups) were collected 90 min after LPS injection, followed by analysis of cytokine secretion. The graphs for several selected cytokines are shown in Fig S5B and D, and all data from Bioplex assays are presented in Table S2.

Both pre and posttreatments with P7-Pen significantly reduced serum levels of IFNβ and IL-12p40 (Fig S5B and D), whereas TNF secretion was not significantly affected at this time point for both treatment protocols, with some trend for the reduction by P7 in the pretreatment group (Fig S5B and D). Interestingly, the level of the anti-inflammatory cytokine IL-10 was significantly increased by 5 nmol/g P7-Pen in both pre and posttreatment groups, with a similar trend for lower peptide concentration in the pretreatment group, which potentially contributed to the anti-inflammatory effect of P7 (Fig S5B and D). The levels of IFNβ and IL-12p40 were among the highest for the mice that died from LPS shock in both water-treated control groups, whereas the levels of TNF or IL-10 did not follow the same trend. The values for dead mice are marked in red on the graphs shown in Fig S5B and D.

Altogether, our results demonstrate that P7 improves the survival and the recovery of animals in murine LPS shock model, which could be mediated by the inhibitory effect of P7 on IFNβ and IL-12p40 secretion. We suggest that P7-Pen could be effective in preventing animal death when administered not only before but also after LPS challenge (Fig S5D). However, additional experiments would be required for the final proof.

## Discussion

In this study, we explored the possibility of blocking TLR4-mediated signaling using a SLAMF1-derived peptide. This peptide was designed based on our previous knowledge of SLAMF1–TRAM interaction domains and finding that SLAMF1 protein positively regulates TLR4-mediated signaling in human monocytes and macrophages (19). We narrowed down the active peptide sequence to 10 amino acids and selected penetratin as an optimal CPP for the intracellular delivery of the lead P7 peptide. We then tested the ability of P7 to inhibit TLR4-mediated signaling in THP-1 macrophage-like cell line and in primary human monocytes and established its strong inhibitory effect on TLR4-mediated signaling, but not on TLR2- or STING-mediated signaling. The P7 peptide significantly inhibited TLR4-mediated mRNA

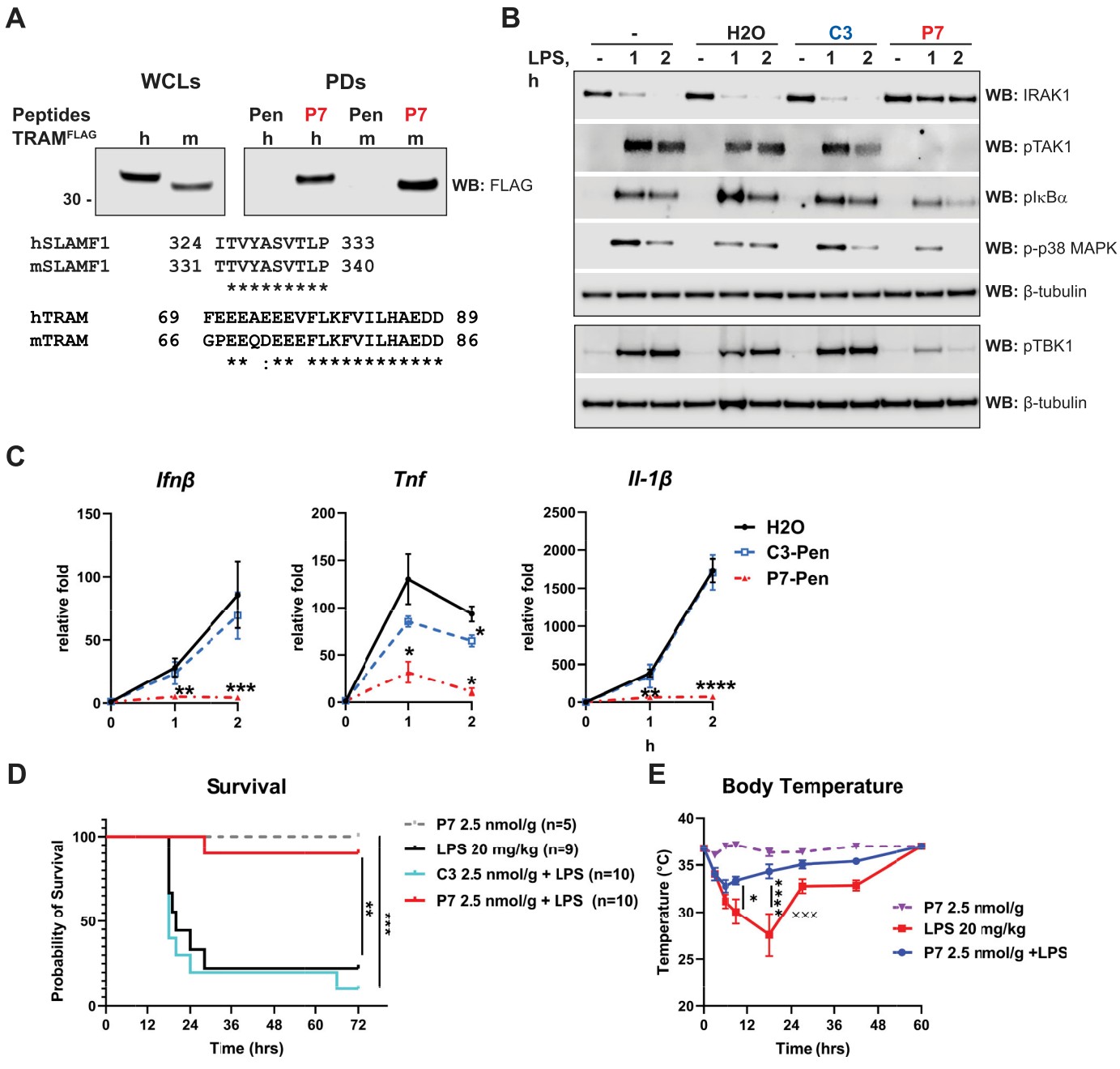

**Figure 9.  P7 interacts with murine TRAM protein, inhibits TLR4-mediated signaling and cytokine mRNA expression in murine macrophages and protects mice from lethal endotoxemia.**

**(A)** HEK 293T cells were transfected by plasmids coding for human or murine TRAM$^{FLAG}$ proteins for 48 h, followed by lysing of cells and PDs by biotinylated peptides Pen or P7-Pen (P7) fixed on NeutrAvidin beads for 1 h. Co-precipitation of FLAG-tagged TRAM orthologs with peptides was addressed by anti-FLAG WB, with WCLs for input control, which represents 14.5% from the total sample used for PD. Sequence alignments for human and murine orthologs of SLAMF1 (in the domain used for peptide design) and TRAM (domain involved in interaction with SLAMF1) are shown below the WB panels. **(B)** Murine immortalized bone marrow-derived macrophages B6 WT were pretreated with 10 μM peptides or a similar amount of sterile water (solvent, $H_2O$) for 30 min and stimulated by LPS (100 ng/ml) for the indicated time. Lysates were analyzed by WB to address TLR4-mediated phosphorylation of TAK1, IκBα, p38 MAPK, TBK1 or posttranslational modifications of IRAK1 (followed by disappearance of the 80-kD band), and β-tubulin WB was used for loading control. **(B, C)** Quantification of *Ifnβ*, *Tnf*, and *Il-1β* mRNA expression by qRT-PCR in B6 WT cells pretreated by peptides or solvent and stimulated by LPS as in (B). Data presented as fold change when compared with unstimulated sample pretreated with water, mean relative fold change ± SD (n = 3). Statistical testing was done by two-way RM-ANOVA. **(D)** C57BL/6J mice were injected i.p. with repurified Sigma smooth *E.coli* 0111:B4 LPS (20 μg/g) or PBS. Peptides (2.5 nmol/g of animal weight, 8.34 or 8.5 mg/kg for C3-Pen and P7-Pen, respectively) were injected i.p. 1 h before injection of LPS. **(D, E)** Probability of survival is shown on (D), and body temperature measurements are shown on (E). Each "x" indicates a succumbed mouse at the time point. **(D, E)** Statistical significance evaluated by log-rank (Mantel–Cox) test with Gehan–Breslow–Wilxocon test (D) or two-way ANOVA for (E). For all graphs, *P < 0.05, **P < 0.01, ***P < 0.001, ****P < 0.0001.
Source data are available for this figure.

expression and secretion of IFN$\beta$ and pro-inflammatory cytokines TNF, IL-1$\beta$, and IL-6 in THP-1 cells and human monocytes.

In general, peptides can be quite unstable, and their efficacy may be diminished in the bloodstream because of binding to plasma proteins or protease cleavage (33). Therefore, we decided to test the P7 peptide in an ex vivo human whole-blood model, which allows for evaluation of the efficacy of P7 inhibition of TLR4-mediated cytokine while maintaining the interplay between all cell types and proteins in blood, within a 5–6-h timeframe. The whole-blood model includes the direct thrombin inhibitor lepirudin as an anticoagulant and has been shown to preserve the active complement system (34). Indeed, we demonstrated that P7 peptide preserved its inhibitory activity towards the secretion of IFN$\beta$, IL-1$\beta$, TNF, IL-6, and MIP-1$\alpha$/CCL3 when both LPS and *E. coli* particles were used for stimulation of blood samples. The inhibitory effect of P7 was concentration-dependent, whereas the control peptide (scrambled) was not inhibiting TLR4-mediated cytokine secretion at any tested concentration. Thus, P7-Pen has high potential for retaining its anti-inflammatory activity in human blood.

When designing the SLAMF1-derived peptide, we primarily aimed to block PPI between SLAMF1 and TRAM, which should, in turn, inhibit TRAM trafficking to the TLR4-containing phagosome and decrease downstream IFN$\beta$ expression and secretion. We have shown that TLR4 and TRAM or TRAM and TRIF PPIs were not affected by the P7 peptide. Based on our results, we hypothesize that P7 is very effective in taking down TLR4-mediated IFN$\beta$ expression and secretion because it targets the formation of the TLR4–TRAM–TRIF complex on several levels. First, by inhibiting the complement-independent TLR4 phagocytosis that is regulated by the interaction of TRAM with Rab11-interacting protein FIP2 and is crucial for TRAM recruitment to TLR4 and phagocytosis (38), as we show here, P7 significantly reduces FIP2 recruitment to bacterial particles in human monocytes. Our data also demonstrated that P7 had only a slight inhibitory effect on complement-driven phagocytosis. Thus, in physiological settings, where bacteria are opsonized by an active complement system, P7 may not have an apparent effect on phagocytosis of bacteria.

Second, TRAM recruitment to bacterial phagosomes after phagocytosis was also significantly inhibited by P7. These effects are likely to be mediated by the inhibition of interaction between FIP2 and TRAM, and SLAMF1 and TRAM, which are both required for TRAM recruitment to endocytosed TLR4 on phagosomes (19, 38).

Third, we received clear indications that the P7 peptide has a target in the TLR4–Myddosome complex, which would explain the inhibitory effect of P7 on the expression and secretion of several pro-inflammatory cytokines (IL-1$\beta$, IL-6, TNF). The expression of the *IFN$\beta$* gene requires the assembly of a transcriptional complex on *IFN$\beta$* gene promoter, which includes IRF3 (activated by TLR4–TRAM–TRIF signaling axis), NF-$\kappa$B, and AP-1 (activated by the TLR4–TIRAP–MyD88 signaling axis) transcriptional factors (50). Thus, blocking of Myddosome formation by P7 results in less MyD88-dependent transcriptional factor activation, which additionally contributes to the inhibition of *IFN$\beta$* expression.

In this study, we found that the SLAMF1-derived P7 peptide blocks TIRAP and MyD88 interaction, whereas not affecting the interaction between TLR4 and TIRAP. These findings were tested in several assays, including cell-free interaction assays with GST-recombinant proteins and overexpression in HEK 293T cells. Interestingly, even though TIRAP is shown to be a bridging adaptor for both TLR2 and TLR4 (45, 48), P7 did not inhibit TLR2-mediated signaling. This could be explained by either the ability of TLR2 receptor to directly recruit MyD88, or by the lower requirement for TIRAP to recruit MyD88 to TLR2, as reported before (49), and demonstrated in the current study. Based on the results from murine in vivo LPS shock model performed in this study, P7 is stable enough to be effective in vivo, significantly preventing LPS-mediated animal death, whereas exhibiting no toxicity on its own. P7 significantly reduced IFN$\beta$ and IL-12p40 plasma levels in both pre and posttreated animals. Overall, these results indicate that P7 has therapeutic potential in TLR4-driven inflammatory conditions.

Although preclinical data obtained from murine models are promising, these animal models fail to account for the genetic, physiological, and immunological differences between mice and humans (51, 52). Therefore, it is important that in addition to murine in vivo models, P7 was tested in human cells, demonstrating efficient inhibition and accounting for human-specific regulatory mechanisms of protein signaling and trafficking.

Inhibition of TLR4-mediated signaling has received a lot of attention in recent years. Many TLR4-targeting drugs are under development and/or in clinical trials, including TLR4 antagonists and small molecules such as TAK-242 (Resatorvid) that binds to TLR4 and interferes with its interaction with two adaptor molecules, TIRAP and TRAM (53, 54). Promising preclinical studies have been conducted with TIR-domain-derived peptides linked to CPP (13, 16, 17, 55). What are then the distinctive attributes of the P7 peptide? Unlike other already developed inhibitory peptides (13), P7 does not disrupt the interaction of TRAM with TLR4 or TRIF, but rather prevents the trafficking of TRAM to TLR4, which in turn depends on the interaction of TRAM with SLAMF1 and FIP2 (19, 38). P7 targets the SLAMF1- and FIP2-interacting motif in N-terminus of TRAM-TIR (69–95 amino acids in humans TRAM), a region that exhibits low homology to other TIR-domain-containing proteins (referred to as region 2 in reference 13) and is required for interaction with FIP2 and SLAMF1 proteins (19, 38).

Similar to certain TIR-domain-containing peptides such as 9R34-ΔN, MyD88-BB, 2R9, 4R9, 6R9, and 9R11 (as reviewed in reference 13), the interaction between the P7 peptide and TIRAP occurs within the TIRAP–TIR domain (amino acids 87–160). Simultaneously, the P7 peptide disrupts TIRAP's interaction with MyD88, yet it does not affect TIRAP's interaction with TLR4 nor does it inhibit TLR2-mediated signaling. This unique combination of effects sets P7 apart from most of the aforementioned TIR-derived peptides with differing inhibitory capabilities. Overall, it is unlikely that a single drug will cure all TLR4-driven pathologies. It is undoubtedly positive that several research groups have reported promising preclinical data with small molecules or TIR domain-derived peptide drug candidates, targeting different branches of PRR signaling and exhibiting different anti-inflammatory effects (14, 15, 16, 17). In the future, such a repertoire of drugs could potentially be of great value to clinicians, enabling the selection of appropriate drugs for different subgroups of patients. The SLAMF1-derived peptide P7 could be important for increasing this repertoire as a drug candidate with its unique mechanism of action and with a

unique effect on cells compared with other peptides in development. Furthermore, as we found that single amino acid substitutions in P7 and other peptide variants of the original sequence could alter its function, with some substitutions being less critical, P7 peptide could be improved, stabilized, and tested for the treatment of many diseases where TLR4-mediated signaling is beneficial to reduce.

# Materials and Methods

## Primary cells

Human buffy coats and serum were from the blood bank at St. Olavs Hospital (Trondheim, Norway), with approval by the Regional Committee for Medical and Health Research Ethics (REC) in Central Norway. Primary human monocytes were isolated from the buffy coat by adherence, as previously described (56). In brief, freshly prepared buffy coats (St. Olavs Hospital) were diluted by 100 ml of PBS and applied on top of Lymphoprep (Axis-Shield) according to the manufacturer's instructions. PBMCs were collected and washed by HBSS (Sigma-Aldrich, Merck) four times with low-speed centrifugation (150–200$g$). Cells were counted using Z2 Coulter particle count and size analyzer (Beckman Coulter) on program B, resuspended in RPMI 1640 (Sigma-Aldrich, Merck) supplemented with 5% of pooled human serum at a concentration of $8 \times 10^6$ per ml, and seeded to six-well (1 ml per well) or 24-well (0.5 ml per well) cell culture dishes. After a 45-min incubation allowing surface adherence of monocytes, the dishes were washed three times by HBSS to remove nonadherent cells. Monocytes were kept overnight in RPMI1640 (Sigma-Aldrich, Merck), supplemented with 30% of pooled human serum, followed by media change to RPMI1640 with 10% human serum before experimental procedures.

## Cell lines

HEK 293T cells (ATCC) were cultured in DMEM with 10% FCS, 100 U/ml penicillin, 100 $\mu$g/ml streptomycin. HEK-Blue IL-1R cells (Invivogen) were cultured in DMEM with 10% FCS, 100 U/ml penicillin, 100 $\mu$g/ml streptomycin. THP-1 WT (monocytic cell line derived from acute monocytic leukemia ATCC TIB-202), and THP-1 sublines THP-1 TRAM-CHERRY (described in reference 38), THP-1 TLR4FLAG (preparation described below) were cultured in RMPI 1640 supplemented by 10% heat-inactivated FCS, 100 U/ml penicillin, 100 $\mu$g/ml streptomycin (Thermo Fisher Scientific), and 5 $\mu$M $\beta$-mercaptoethanol (Sigma-Aldrich, Merck). For preparation of THP-1 TLR4FLAG cells, TLR4 with C-terminal FLAG tag (DYKDDDDK) synthetic construct (ATG:bio-synthetics) was inserted to pLVX-EF1$\alpha$-IRES-puro vector (Clontech, Takara Bio USA). Coding vector was co-transfected with packaging plasmids psPAX2 and pMD2.G (kindly provided by the Trono Lab, plasmids #12260 and #12259; Addgene) to HEK 293T cells to produce pseudoviral particles. Supernatants were collected at 48 h and concentrated using Lenti-X Concentrator (Clontech, Takara Bio USA) and used for transduction of THP-1 WT cells, followed by selection on puromycin (1 $\mu$g/ml) for 3 wk. Prior experimental procedures THP-1 WT, THP-1 TLR4FLAG or THP-1 TRAMCHERRY cells were differentiated with 60 ng/ml of phorbol 12-myristate 13-acetate (PMA) (Sigma-Aldrich, Merck) for 48 h, followed by 48 h in a medium without PMA. Media were changed to fresh media before the pretreatments and stimulation.

## Reagents and cell stimulation

Synthetic peptides for assays with living cells were from GenSript, Innovagen, or Thermo Fisher Scientific. Peptide modifications and characteristics: N-terminal acetylation, C-terminal amidation, >90% purity, guaranteed TFA removal, control for endotoxin levels (less then < 10 EU/mg). Biotinylated peptides were from GenSript: >95% purity, N-terminal acetylation, C-terminal biotinylation (to C-terminal lysin of penetratin), TFA salts. pHrodo red and AF488 labeled $E.$ $coli$ bioparticles were purchased from Thermo Fisher Scientific and reconstituted in 2 ml PBS ($3 \times 10^8$/ml). Particles used for different assays: $1 \times 10^6$/ml for the whole blood assays, 3 particles/cell for experiments analyzed by confocal microscopy, 4–5 particles/cell for flow cytometry phagocytosis assays. Ultrapure K12 LPS from $E.$ $coli$, synthetic diacylated lipoprotein FSL-1 (Pam2CGDPKHPKSF), 2′3′-cGAMP VacciGrade. For stimulation of the primary cells and THP-1 cells, LPS and FSL-1 were used at a concentration of 100 ng/ml, 2′3′-cGAMP —20 $\mu$g/ml. Complement inhibitor compstatin (43) (a kind gift from Dr. JD Lambris and Dr. TE Mollnes) was used in 20 $\mu$M concentration for bacterial particle opsonization mixture. HEK-Blue IL-1R cells were seeded to six-well plates ($3 \times 10^5$/well) in fresh media and in 24 h stimulated with human recombinant IL-1$\beta$ (20 ng/ml) from R&D Systems (Biotechne).

## Antibodies

The following primary antibodies were used for Western blotting: rabbit anti–TICAM-2/TRAM (GTX112785) from Genetex; rabbit mAb anti–human SLAMF1 (10837-R008-50) from Sino Biological Inc.; rabbit $\beta$-tubulin (ab6046) from Abcam; phospho-p38 MAPK (T180/Y182), phospho-STAT1 (Tyr701) (58D6), phospho-TAK1 (T184/187) (90C7), IRAK1 (D51G7), MyD88 (D80F5), phospho–IkB-$\alpha$ (14D4), phospho-TBK1/NAK (Ser172; D52C2), anti-DYKDDDDK tag (D6W5B)/Flag tag from Cell Signaling Technology; Living Colors rabbit anti–Full-Length GFP polyclonal antibodies (632592) from Takara Bio Inc.; rabbit PCNA Abs were from Santa Cruz Biotech (Santa Cruz); sheep IRAK1 (used for IPs) and IRAK4 (Used for WB and IPs) were from MRC-PPU Reagents (University of Dundee, Dundee, UK); rabbit anti-HA tag polyclonal Abs (#71-5500), goat TIRAP polyclonal Abs (#PA5-18439) were from Invitrogen. Secondary antibodies (HRP-linked) were from DAKO Denmark A/S. Antibodies used for confocal microscopy—rabbit anti–TICAM-2 from Genetex, recombinant rabbit anti-RAB11-FIP2 antibody (EPR12294-85) (ab180504) from Abcam. Secondary antibodies for confocal microscopy were Alexa Fluor 647 conjugate (A-21235) from Thermo Fisher Scientific.

## Expression vectors, cloning

SLAMF1 in pcDNA3.1 for expression of SLAMF1 WT, SLAMF1 C-terminal DYKDDDDK for SLAMF1FLAG (WT), TRAM C-terminal DYKDDDDK for TRAMFLAG, TLR4 C-terminal DYKDDDDK for TLR4FLAG preparation described in reference 19, TRIFHA, TRAMYFP, MyD88CFP, TIRAPHA, TLR2FLAG coding constructs from K. Fitzgerald (University of Massachusetts Medical School, Worcester, MA). Rab11FIP2 in pEGFPC1

vector was from M McCaffrey (University College Cork, Cork, Ireland). SLAMF1 P333T mutant with FLAG-tag was prepared by subcloning of SLAMF1 WT to C-terminal DYKDDDDK vector (EcoRI-XhoI) using reverse primer coding for mutation (For 5′-TTAGAATTCATGGATCCCAAGGGGCTCC-3′, Rev 5′- TATCTCGAGGCTCTCACGAAGTGTCACACTAGCA-3′) TIRAP was subcloned to C-terminal DYKDDDDK (Clontech) vector (EcoRI-XhoI) using primers (For 5′- TATGAATTCATGGCATCATCGACCTC-3′, Rev 5′- TACCTCGAGAGCCGGTACTGAGTGTCTGCAG-3′). For subcloning of partial TRAM and TIRAP coding sequences to pGEX-2TK vector (GE Healthcare) for GST-fusion protein production in bacteria, the following primers were used: GST-TRAM (2–100 aa)—for 5′- ATAGGATCCGGTATCGGGAAGTCTAAAATAAATTC-3′, Rev ATGAATTCTCATAGCAGATTCTGGACTCTG -3′; GAT-TIRAP (87–160 aa) For 5′-TATGGATCCGACGTCTGCGTGTGCCAC-3′, Rev 5′- TAT-GAATTCTCACTGGTACTTGCACCAGGGGTC-3′; GST-TIRAP (30–74 aa) For 5′- TATGGATCCCTGAAGAAGCCCAAGAAGAGG-3′, Rev 5′- TAT-GAATTCTCACGCATGTGTGGGTGGCAG-3′. Phusion high-fidelity DNA polymerase and respective Fast Digest enzymes (Thermo Fisher Scientific) were used for re-cloning. Plasmids were purified by the Endofree plasmid maxi kit (QIAGEN). Sequencing of plasmids was done at the Eurofins genomics facility.

## qRT-PCR

Total RNA was isolated from the cells using Qiazol reagent (QIAGEN), and chloroform extraction was followed by purification on RNeasy Mini columns with DNAse digestion step (QIAGEN). cDNA was prepared with a Maxima First Strand cDNA Synthesis Kit (Thermo Fisher Scientific), in accordance with the protocol of the manufacturer using 400–600 ng of total RNA per sample. qPCR was performed with the PerfeCTa qPCR FastMix (Quanta Biosciences) in replicates and cycled in a StepOnePlus Real-Time PCR cycler (Thermo Fisher Scientific). The following TaqMan Gene Expression Assays (Applied Biosystems, Thermo Fisher Scientific) were used: *IFNβ* (Hs01077958_s1), *TNF* (Hs00174128_m1), *TBP* (Hs00427620_m1), *IL-6* (Hs00985639_m1), *IL-1β* (Hs01555410_m1) for human cells; *Ifnβ* (Mm00439552_s1), *Tnf* (Mm00443258_m1), *Il-1β* (Mm00434228_m1), and *Tbp* (Mm01277042_m1) for murine cells. The level of *TBP* mRNA was used for normalization and the results presented as a relative expression compared with the control's untreated sample. Relative expression was calculated using Pfaffl's mathematical model (57). Graphs and statistical analyses were made with GraphPad Prism v9.1.2 (Dotmatics), with additional details provided in the Figure legends and statistics paragraph.

## ELISA, Bio-Plex, and LDH release assay

IFNβ levels were determined using assays from PBL Assay Science: VeriKine-HSTM Human Interferon-Beta Serum ELISA Kit (#41415) for human cells' supernatants or plasma samples, and VeriKine-HS mouse IFNβ serum ELISA kit (#42410-1) for murine plasma samples. Other cytokines (TNF, IL-1β, IL-6, MCP-1α/CCL3, IL-10, IP-10/CXCL10) for human primary cells and plasma samples and 23 cytokines for murine cells (#M60009RDPD; 23-plex Assay) were analyzed using BioPlex cytokine assays from Bio-Rad, in accordance with the instructions of the manufacturer, using the Bio-Plex Pro Reagent Kit III and Bio-Plex 200 System (Bio-Rad). For the analysis of cytokines

secretion by THP-1 cells, PMA differentiated THP-1 cells were pretreated with solvents or peptides (12.5 μM) for 30 min followed by 4 h of stimulation with UP K12 LPS (100 ng/ml) and analysis of cytokine expressions in supernatants. All screens were performed in three to five biological replicates for each treatment/peptide. The following ELISA assays were used: human CXCL10/IP-10 DuoSet ELISA (DY266) and human TNF DuoSet ELISA (DY210) from R&D Systems (Biotechne) and human IL-1β ELISA Set II kit (BD OptEIA, BD Biosciences). CyQUANT LDH Cytotoxicity Assay (Thermo Fisher Scientific) was used to measure the extracellular LDH in supernatants as suggested by the manufacturer.

## Western blotting

Cell lysates for Western blotting (WB) analysis when indicated were prepared by simultaneous extraction of proteins and total RNA using Qiazol reagent (QIAGEN) as suggested by the manufacturer. Extracted total RNA was used for qRT–PCR, whereas protein samples were used for simultaneous analysis of protein expression/posttranslational modifications. Protein pellets were dissolved by heating the samples for 10 min at 95°C in a buffer containing 4 M urea, 1% SDS (Sigma-Aldrich, Merck), and NuPAGE LDS Sample Buffer (4X) (Thermo Fisher Scientific), with a final 25 mM DTT in the samples. Otherwise, lysates were made using 1X RIPA lysis buffer: 150 mM NaCl, 50 mM Tris–HCl, pH 7.5, 1% Triton X-100, 5 mM EDTA, supplemented with EDTA-free Complete mini protease inhibitor cocktail tablets and PhosSTOP phosphatase inhibitor cocktail (Roche), 50 mM NaF, and 2 mM $Na_3VO_3$ (Sigma-Aldrich). For SDS–PAGE, we used pre-cast gradient 4–12% Bis-Tris protein gels NuPAGE Novex and either MOPS (for resolution of 25–200 kD proteins) or MES (for resolution of 2–60 kD peptides/proteins) SDS running buffer (20X) (Thermo Fisher Scientific), and proteins from gel were transferred to iBlot Transfer Stacks by using the iBlot Gel Transfer Device (Thermo Fisher Scientific). The blots were developed with the SuperSignal West Femto (Thermo Fisher Scientific) and visualized with the LI-COR ODYSSEY Fc Imaging System (LI-COR Biotechnology). For densitometry analysis of the WB bands, Odyssey Image Studio 5.2 software (LI-COR Biotechnology) was used, and the relative numbers of bands' intensity were normalized to the intensities of the respective loading-control protein (β-tubulin or PCNA).

## Whole blood assay

Blood samples were obtained from healthy volunteers that gave signed consent for experimental procedures, approved by REC in Central Norway (REK#S-04114). Refludan (lepirudin) was used as the anticoagulant in 50 μg/ml concentration as described before (34). Blood samples were distributed to sterile polypropylene tubes, 0.25 ml per sample, with the total volume of reagents added to each sample being 0.1 ml (in PBS). After addition of peptides (10, 20 or 40 μM) or solvent (water, $H_2O$) for 30 min, samples were stimulated with LPS (100 ng/ml) or $1 × 10^6$ of pHrodo red *E. coli* bioparticles for 5 h at 37°C (while shaking 400 rpm). Blood was transferred to tubes containing ethylenediamine tetra acetic acid (EDTA, 10 mM), spun down at 3,220*g* at 4°C for 15 min for plasma separation, and the plasma frozen for later cytokine expression analysis.

## Immunoprecipitations

PBMC-derived monocytes for endogenous IPs were lysed using 1 X lysis buffer (150 mM NaCl, 50 mM Tris–HCl [pH 8.0], 1 mM EDTA, 1% NP40) and supplemented with EDTA-free Complete Mini protease Inhibitor Cocktail Tablets and a PhosSTOP phosphatase-inhibitor cocktail from Roche, with 50 mM NaF and 2 mM $Na_3VO_3$ (Sigma-Aldrich, Merck). Protein concentration in lysates was established using Pierce BCA Protein Assay kit (Thermo Fisher Scientific). Immunoprecipitations (IPs) were carried out on rotator at +4°C for 4 h by co-incubation of the lysates from the stimulated cells (500 $\mu$g of protein/IP) with polyclonal anti-IRAK1, anti-IRAK4 (sheep) anti-TIRAP (goat) antibodies covalently coupled to Dynabeads (M-270 Epoxy; Thermo Fisher Scientific) as suggested by the manufacturer, followed by extensive washing of the beads with cold lysis buffer (no inhibitors) and elution of co-precipitated proteins. For anti-FLAG IPs, HEK 293T cells were transfected with constructs coding FLAG-tagged, and HA-, EGFP-, EYFP- or ECFP-tagged or untagged proteins, in six-well plates, using 0.2–0.4 $\mu$g of vector/well and 3 $\mu$l/well of GeneJuice Transfection Reagent (EMD Millipore). After 48 h, cells were pretreated by peptides if indicated in Figure legends, washed with PBS, and harvested in lysis buffer prepared as described for endogenous IPs. IPs were performed using 35 $\mu$l/IP of anti-FLAG M2 affinity agarose solution (Sigma-Aldrich, Merck), beads washed by lysis buffer before adding respective lysates. IPs were carried out on a rotator at +4°C for 4 h. Co-precipitated complexes from endogenous IPs and anti-FLAG IPs were eluted by heating the samples in a 1× loading buffer (LDS, Invitrogen, and Thermo Fisher Scientific) without addition of a reducing agent to minimize the antibodies' leakage to the eluates. Eluates were transferred to clean tubes, DTT (Sigma-Aldrich, Merck) added to 40 mM concentration, samples heated, and analyzed by SDS–PAGE and WB. Precipitates were loaded to the gels in parallel with respective whole-cell lysates for input control.

## Preparation of GST proteins for the pulldown assays

Empty pGEX-2TK vector (for preparation of GST protein) and vectors coding for GST-TRAM (2–100 aa), GST-TIRAP (87–160 aa), GST-TIRAP (30–74 aa) were transformed into the BL21 DE3 bacterial strain (New England Biolabs). For protein purification, we used Pierce Glutathione Agarose (Thermo Fisher Scientific). Expression and purification of GST fusion proteins were performed as described previously (58). GST and GST-tagged proteins were eluted from beads by the elution buffer (10 mM reduced glutathione [Sigma-Aldrich, Merck] in 50 mM Tris–HCl [Sigma-Aldrich, Merck], pH 8.0), buffer exchanged to PBS using dialysis columns centrifugal filter units Amicon Ultra 3K from Merck Millipore (Merck). Protein concentration evaluated by spectrophotometry.

## Pull down assays by biotinylated peptides

For pull downs (PDs) by biotinylated peptides, 30 $\mu$l/PD of Pierce NeutrAvidin Agarose beads (Thermo Fisher Scientific) were washed by PBS and coated with biotinylated peptides (2.5 nmol/PD) for 30 min at RT, on a rotator. Beads were washed by PD assay buffer twice and distributed to the respective amount of tubes for PDs. Cell lysates of untreated or LPS-stimulated monocytes for pulldowns were prepared in 1×lysis buffer used for endogenous IPs (described in section on immunoprecipitations). Protein concentration in lysates was measured using Pierce BCA Protein Assay Kit (Thermo Fisher Scientific), and 500 $\mu$g of total protein used for each PD with peptide-coated beads. Precipitation was performed at the rotator, +4°C for 30 min. Beads were washed with ice-cold 1×lysis buffer, followed by addition of 1× loading buffer (LDS, Invitrogen, and Thermo Fisher Scientific) containing 40 mM DTT (Sigma-Aldrich, Merck), followed by heating of samples, SDS–PAGE and WB analysis. For PD with purified GST-tagged proteins, agarose beads, coated by peptides as described in the beginning of this section, were washed with a cold assay buffer containing 400 mM NaCl, 50 mM Tris–HCl, pH 7.5, 1% Triton X-100, 5 mM EDTA, followed by addition of 300 $\mu$l of the assay buffer and 1 $\mu$g of recombinant protein per PD. PDs were gently mixed and placed on the rotator for 15 min (at room temperature), washed with assay buffer, and co-precipitated proteins and peptides eluted by heating of beads with 1xLDS, 40 mM DTT. After SDS–PAGE, gels were stained by SimplyBlue Safe Stain (Thermo Fisher Scientific) as suggested by the manufacturer and washed by water to reduce the background staining for 2 h. Images were taken on Gel Doc EZ Imager (Bio-Rad, Bio-Rad Laboratories) and densitometry analysis of stained bands performed using Image Lab Software 6.0.1 (Bio-Rad).

## Phagocytosis flow cytometry assays and particle opsonization

A flow cytometry-based phagocytic assay was used to measure the phagocytic efficiency of red pHrodo-conjugated *E. coli* BioParticles in primary human monocytes. According to the manufacturer, the pHrodo dye conjugates are low-fluorescent outside the cell but fluoresce brightly in phagosomes after uptake. Before being added to cells, the bacterial bioparticles were either not opsonized and diluted in stimulation media just before addition to the cells or opsonized with different settings. For opsonization, bacterial particles where diluted to 10-fold higher concentration that was used for stimulating the cells in media containing the following: (1) 30% normal pooled human A$^+$ serum (stored frozen in aliquots to preserve complement system activity); (2) 30% human serum with addition of 20 $\mu$M complement inhibitor compstatin that binds to C3 and inhibits proteolytic cleavage by C3 convertase and activation of classical and alternative complement pathways (43); (3) 30% human heat-inactivated serum. Particles were placed on water bath (37°C) for 15–20 min and added to the cells for the indicated time. After stimulation, cells in six-well plates were placed on ice, washed with cold PBS, detached by treatment with Accutase solution for 10–15 min (Sigma-Aldrich, Merck), and transferred into FACS tubes. The cells were washed with PBS containing 2% FCS followed by PBS. The fluorescence intensity was measured with a BD LRSII flow cytometer using the FACS Diva software (BD Biosciences). Data were exported and analyzed with FlowJo software v10.0.5 (Tree Star).

### Immunostaining and confocal microscopy

Confocal images were captured using TCS SP8 (Leica Microsystems) equipped with a high-contrast Plan Apochromat 63 × 1.40 NA oil CS2 objective. Acquisition software used is LAS AF software (4.0.0.11706; Leica Microsystems). Before imaging, cells were fixed on ice with 4% paraformaldehyde in PBS. Immunostaining was performed as previously described (56). In brief, upon fixation, the cells were permeabilized with PEM buffer (80 mM K-Pipes, pH 6.8, 5 mM EGTA, 1 mM $MgCl_2$, 0.05% saponin) for 15 min on ice, quenched of free-aldehyde groups in 50 mM $NH_4Cl$, 0.05% saponin for 5 min, and blocked in PBS with 20% human serum and 0.05% saponin. The cells were incubated with primary antibody in PBS with 2% human serum and 0.05% saponin overnight at 4°C. After three washes in PBS with 0.05% saponin, secondary Abs were incubated for 15 min at RT, followed again by three washes. Images of stained cells were captured at RT, and 3D data were captured with identical settings, which were adjusted to avoid saturation of voxel (3D pixels) intensities. The AF488 fluorescence was used to spot or surface render the volume of individual phagosome AF488-conjugated *E. coli* particles. For this, a binary mask was created around the bacterial particles (Process/Make Binary function) and used to define the regions for quantification of mean intensities (Mis) for TRAM and FIP2 voxels in original images when redirected to the original image and to quantify *E. coli* AF488 particles number per cell. Statistical significance evaluation is described in the statistics paragraph and briefly in Figure legends.

### Animal studies

Animal experiments were performed with approval of UMass Institutional Animal Care and Use Committee. C57BL/6J mice originally from Jackson Laboratory were bred at UMass animal facility. Gender-matched mice with the age of 8–12 wk were used in each experiment. For the LPS shock model, repurified TLR2 ligand-free LPS (*E. coli* O111:B4; Sigma-Aldrich, Merck) diluted in PBS was intraperitoneally (i.p.) injected at 20 mg/kg body weight. Control group received the same amount of PBS. C3 and P7 peptides were reconstituted in tissue culture-grade water and injected i.p. 1 h before LPS challenge (2.5 nmol/g of animal weight, 8.34 or 8.5 mg/kg for C3-Pen and P7-Pen, respectively). Body condition score and temperature were monitored for 72 h after LPS injection. Rectal body temperature was measured by a thermocouple thermometer (Digi-Sense) with a rectal probe (Physitemp RET3; Physitemp Instruments LLC) at 3, 6, 9, 18, 27, 42, and 60 h post LPS injection. Animals with body temperature below 25C or a body condition score less than two out of five-point scale were identified as humane endpoints and euthanized.

An animal study shown in supplementary files included both post and pretreatment groups and was run by Melior Discovery Inc.. The animal procedure was conducted according to the established protocols by the IACUC and Melior Standard Operation Procedures. Study included C57BL/6J 6–7 wk healthy male mice (Charles River Labs) (n = 69), acclimated for 1 wk, housed on a 12-h light/dark cycle. Peptide (water solution) or water and LPS were injected i.p. Repurified Sigma smooth *E. coli* 0111:B4 LPS or PBS (for the control untreated group) was injected i.p. at 20 mg/kg (at volume 4 ml/kg).

Study included six animal groups–control group (untreated) (n = 12), pretreatment by water (vehicle) (n = 12), pretreatment by 2.5 nmol/g or 8.5 mg/kg P7-Pen (n = 11), pretreatment by 5 nmol/g or 17 mg/kg P7-Pen (n = 11), posttreatment by water (vehicle) (n = 12), and posttreatment by 5 nmol/g P7-Pen (n = 11). Body weight was monitored daily. Mortality was registered for the nearest hour from death, temperature was evaluated at 2, 4, 6, 10, 20, 24, 30, and 48 h from injection of LPS using infrared thermometer, blood samples were collected once at 90 min after LPS injection by retroorbital eye bleeding into EDTA tubes for plasma preparation.

### Statistical analysis

Data that were assumed to follow a log-normal distribution was log-transformed before statistical analysis. Quantification of genes expression qRT-PCR was log-transformed and analyzed by Repeated Measures Analysis of Variance (RM-ANOVA), or a mixed model if there was missing data, followed by Holm-Šídák's multiple comparisons post-test. ELISA and BioPlex data were analyzed using a Wilcoxon matched-pairs signed-rank test or Mann–Whitney test. For the data from WB analysis of IPs, significance was evaluated by *t* test with Welch's correction. For the data of quantification of MVI from confocal microscopy, statistical significance was evaluated by two-way ANOVA and for the quantification of the uptake of bacterial particles using one-way ANOVA. Statistical significance for quantification of flow cytometry bacteria uptake assays was evaluated by Mann–Whitney test in pairwise comparison with control treatment. For the pull-down assays with GST-fusion proteins, statistical significance was evaluated by one-way ANOVA with Dunnett's multiple comparison test. For the survival analysis in murine LPS shock/endotoxemia model significance evaluated by Log-rank (Mantel–Cox) test with Gehan–Breslow–Wilxocon test, and for body temperature evaluation either mixed effects model (REML) or multiple Mann–Whitney test as indicated in the figure legends. Significance for the body weight (BW) change was evaluated using multiple Mann–Whitney test. All graphs and analyses were generated with GraphPad Prism v9.5.0 (Dotmatics).

# Supplementary Information

# Acknowledgements

We want to thank Dr. Lene Grøvdal (NTNU, Norway) for providing THP-1 TRAM[CHERRY] subline, and Dr. JD Lambris (University of Pennsylvania, PA, USA) and Dr. TE Mollnes (University of Oslo, Norway) for providing complement inhibitor compstatin. This research was funded by the Research Council of Norway through its Centers of Excellence Funding Scheme, Grant 223255/F50 (to T Espevik), NTNU Discovery Grant 2020 (to T Espevik and M Yurchenko), the Liaison Committee for Education, Research and Innovation in Central Norway Innovation Researcher Grant 90794301 (to M Yurchenko) and Felles Forskningsutvalg (FFU) Grant 2022/2758 (to M Yurchenko).

## Author Contributions

KE Nilsen: formal analysis, methodology, investigation, and writing—original draft, review, and editing.
B Zhang: formal analysis, investigation, and writing—review and editing.
A Skjesol: investigation.
L Ryan: formal analysis and investigation.
H Vagle: investigation.
MH Bøe: investigation.
P Orning: investigation.
H Kim: investigation.
SS Bakke: investigation.
K Elamurugan: investigation.
IB Mestvedt: investigation.
J Stenvik: investigation and writing—review and editing.
H Husebye: investigation and writing—review and editing.
E Lien: resources, investigation, and methodology.
T Espevik: conceptualization, resources, supervision, funding acquisition, methodology, and writing—review and editing.
M Yurchenko: conceptualization, resources, formal analysis, supervision, funding acquisition, investigation, methodology, project administration, and writing—original draft, review, and editing.

## Conflict of Interest Statement

The authors declare that they have no conflict of interest.

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
