## [Reviewer comments · Life Science Alliance]

Life Science Alliance

Peptide derived from SLAMF1 prevents TLR4-mediated inflammation in vitro and in vivo

Kaja Nilsen, Boyao Zhang, Astrid Skjesol, Liv Ryan, Hilde Vagle, Maren Bøe, Pontus Orning, Hera Kim, Siril Bakke, Kirusika Elamurugan, Ingvild Mestvedt, Jørgen Stenvik, Harald Husebye, Egil Lien, Terje Espevik, and Maria Yurchenko

DOI: <https://doi.org/10.26508/lsa.202302164>

Corresponding author(s): Maria Yurchenko, Norwegian University of Science and Technology

Review Timeline:	Submission Date:	2023-05-16
	Editorial Decision:	2023-06-30
	Revision Received:	2023-08-20
	Editorial Decision:	2023-09-13
	Revision Received:	2023-09-21
	Accepted:	2023-09-22

Scientific Editor: Novella Guidi

Transaction Report:

June 30, 2023

Re: Life Science Alliance manuscript #LSA-2023-02164-T

Dr. Maria Yurchenko
Norwegian University of Science and Technology
Centre for Molecular Inflammation Research (CEMIR)
Olav Kyrres gate 10
NTNU Campus Øya, Kunnskapsenteret, 3rd Floor
Trondheim, Sør-Trøndelag 7491
Norway

Dear Dr. Yurchenko,

Thank you for submitting your manuscript entitled "Peptide derived from SLAMF1 prevents TLR4-mediated inflammation in vitro and in vivo" to Life Science Alliance. The manuscript was assessed by expert reviewers, whose comments are appended to this letter. We invite you to submit a revised manuscript addressing the Reviewer comments.

Thank you for this interesting contribution to Life Science Alliance. We are looking forward to receiving your revised manuscript.

Sincerely,

B. MANUSCRIPT ORGANIZATION AND FORMATTING:

Reviewer #1 (Comments to the Authors (Required)):

Nilsen et al. develop a cell-permeable peptide that inhibits signaling by several TLRs due to binding to and sequestering TIR domains of TIRAP and TRAM. Authors follow up on their previous study that discovered that SLAMF1 can bind TRAM directly and by doing so it facilitates TRAM signaling by helping trafficking the adapter to endosomes. Authors first probed a series of CFP-tagged peptides, each derived from the putative TRAM-interacting site of SLAMF1, for blocking the TRAM-SLAMF1 interaction. Peptide that best inhibited the interaction (named P7) also inhibited (when rendered cell-permeable by conjugation with a peptide vector) the LPS-induced IFN β and (unexpectedly) TNF α in THP1 cells and other human and mouse monocytes. P7 however did not block the TLR2- or STING-induced cytokines but did inhibit the ODN2006-stimulated TLR9 responses. Synthetic P7 inhibited co-IP of TRAM with SLAMF1 and FIP2 but surprisingly not with TLR4 or TRIF. A series of co-IP assays has demonstrated that the peptide blocked TIRAP-MyD88 interaction and MyDDosome assembly. P7 also reduced complement-independent phagocytosis. P7-Pen demonstrated a significant suppression of LPS-induced toxicity manifestations in vivo and improved survival in the mouse model of LPS-induced shock. Authors propose that P7 inhibits the proinflammatory TLR signaling by sequestering TIRAP, whereas phagocytosis and the MyD88-independent arm of TLR4 signaling are inhibited due to P7 binding to SLAMF and FIP2, and consequent suppression of TRAM trafficking to endosomal compartments. Overall, the study is technically sound and valuable, and demonstrates proficiency in TLR studies. I have however a few suggestions/questions, answering of which would make the publication stronger, in my opinion.

The statement that "LPS binding induces homodimerization, causing conformational changes in the intracellular TIR domains..." requires a reference to the primary data.

A 2020 review on cell-permeable peptide inhibitors of TLRs listed ~3 dozen of such inhibitors (and more were discovered since), including peptides that bind TIRAP and TRAM. Authors should discuss how P7 effects are different from the effects of previously reported peptides.

Selection of P7 for further studies was made based on its ability to co-IP with TRAM (Fig 1). Is P7 optimal of the series for inhibition of MyD88-mediated TLR signaling?

Does P7-Pen inhibit the ODN1668-stimulated TLR9? ODN1668 induces proinflammatory cytokines but reportedly does not induce much of IFN response.

Labeling of blots shown in Fig 4 is confusing. Why do peptide treatments so strongly augment TRAM expression in WCLs?

"...opsinized particles for 45 min, flowed by..."

What are the "matching water treatments" for data in water-treated columns in Table S2?

Reviewer #2 (Comments to the Authors (Required)):

The work submitted by Nilsen and colleagues aimed to investigate the inhibitory effects of a peptide, named P7, designed against the signalling lymphocyte activation molecule family 1 (SLAMF1) in the TLR4-TRAM-TRIF signaling pathway. The authors successfully optimized the intracellular delivery of the peptide by linking it to Penetratin (Pen). They demonstrated that P7-Pen specifically inhibits cytokine production in TLR4 signaling triggered by LPS in human primary monocytes. The inhibitory effect of P7 was also observed in a human whole blood model, where it inhibited LPS and bacteria-induced cytokine production. The authors performed co-precipitation studies and showed that P7 inhibits the interaction between SLAMF1 and TRAM, as well as Rab11 FIP2 and TRAM, without affecting TRAM's interaction with TLR4 or TRIF. Furthermore, they demonstrated that P7 inhibits the recruitment of the TRAM-FIP2 complex to phagosomes in THP-1 cells and primary human monocytes. Co-precipitation experiments revealed the interaction of P7 peptide with endogenous TRAM adapter molecule and MyD88-related signaling molecules in primary human monocytes. The authors extended their findings to a mouse model and demonstrated that

P7 peptide interacts with mouse TRAM adapter molecule, resulting in the inhibition of TLR4-mediated signaling and cytokine mRNA expression in mouse immortalized BMMs. Additionally, in an in vivo model of endotoxemia in mice, P7 peptide protected mice from lethal endotoxemia, providing a proof-of-concept for its therapeutic potential.

I find the paper to be well written, and the research is well performed. However, I have identified a few minor points that should be addressed by the authors:

1. In the introduction, please rephrase the statement about the binding of PAMPs (or DAMPs) to the extracellular domain of PRRs, considering that some TLRs may occur both at the cell membrane and intracellularly.
2. For the methods related to Fig.1a and Fig.6b, please indicate the polyacrylamide gel concentration used.
3. In Fig.1a, there is a doublet in the Western blot panel of Flag/TRAM(Flag). Please indicate the band corresponding to overexpressed TRAM.
4. For the lower panels (co-IP results) in Fig.1a, it would be helpful if the authors provide a figure with all samples in the same gel.
5. In Fig.1b 4a and 4b, the images should be adjusted adequately to preserve an adequate amount of background information.
6. For Fig.1C, it would be beneficial to discuss why the authors did not perform a similar analysis for T7-Pen.
7. Provide more details regarding the methodology used for the development of the P7 peptide. Include information about how the single amino acid substitutions or additions were theoretically planned and defined, as well as the synthesis, purification, and characterization of the peptide. This will enhance the reproducibility of the study.
8. Consider removing the data for the effects of the peptide on TLR9 signaling, as it may be unnecessary.
9. In Fig.3A and B, the authors should discuss why human primary monocytes pretreated with P7-Pen and further exposed to bacteria do not reconcile the data on CCL3 and IL-8 when treated with LPS.
10. The text on the beginning of page 10 can be improved for brevity. There is no need to reintroduce the knowledge already covered in the introduction section.
11. In the co-precipitation and pull-down experiments, it would be helpful if the authors provide a Western blot image of a percentage of the input, whenever possible.
12. The first paragraph of the Discussion section seems unnecessary, as the authors have objectively discussed those points in the Introduction section. The text can start from the second paragraph.

Overall, I believe that the paper makes a significant contribution to the field by introducing a novel SLAMF1-derived peptide with a unique mode of action that effectively inhibits TLR4-mediated signaling. The research presented is well-executed, and the results are supported by robust experimental data. I recommend the paper for publication after addressing the minor points mentioned above.

By this letter we want to thank the reviewers for the critical evaluation of our manuscript “Peptide derived from SLAMF1 prevents TLR4-mediated inflammation *in vitro* and *in vivo*”. In this letter we provide point-by-point reply to all the reviewers’ comments.

Answers to Reviewer #1 comments:

1) The statement that "LPS binding induces homodimerization, causing conformational changes in the intracellular TIR domains..." requires a reference to the primary data.

Thank you for this correction, the reference (Bovijn, 2012) was included to the text of the manuscript.

2) A 2020 review on cell-permeable peptide inhibitors of TLRs listed ~3 dozen of such inhibitors (and more were discovered since), including peptides that bind TIRAP and TRAM. Authors should discuss how P7 effects are different from the effects of previously reported peptides.

Thank you for this comment, it is indeed very important to emphasize what are unique and common features of P7 peptide when compared to other peptides developed to inhibit TLRs’ signaling pathways.

The discussion part of the manuscript was edited to include these points (inserted text is highlighted in the manuscript):

Inhibition of TLR4-mediated signaling has received a lot of attention in recent years. Many TLR4-targeting drugs are under development and/or in clinical trials, including TLR4 antagonists and small molecules such as TAK-242 (Resatorvid) that binds to TLR4 and interferes with its interaction with two adaptor molecules, TIRAP and TRAM (52, 53). Promising pre-clinical studies have been conducted with TIR-domain-derived peptides linked to CPP (13, 16, 17, 54). **What are then the distinctive attributes of P7 peptide? To begin with, P7 is not made from the TIR or TIR-like domain. As a result, its capability for multispecific binding to various TIR-domain-containing molecules is unlikely. This is in contrast to a defining characteristic of TIR-derived peptides (13). Several other peptides from microbial or viral origins, unrelated to the TIR domain, were recently described, however, the region they target within TIRAP or TRAM adaptor proteins (as reviewed in (13)) does not overlap with that of P7. Unlike other already developed inhibitory peptides (13), P7 does disrupt interaction of TRAM with TLR4 or TRIF, but rather block TRAM recruitment to TLR4. P7 targets the N-terminus of TRAM-TIR, a region**

that exhibits low homology to other TIR-domain-containing proteins (referred to as region 2 in (13)) and is required for interaction with FIP2 and SLAMF1 proteins and TRAM trafficking ((19, 38)).

Similar to certain TIR-domain-containing peptides such as 9R34-ΔN, MyD88-BB, 2R9, 4R9, 6R9, and 9R11 (as reviewed in (13)), the interaction between the P7 peptide and TIRAP occurs within the TIRAP-TIR domain (amino acids 87-160). Simultaneously, the P7 peptide disrupts TIRAP's interaction with MyD88, yet it does not affect TIRAP's interaction with TLR4 nor does it inhibit TLR2-mediated signaling. This unique combination of effects sets P7 apart from the majority of the aforementioned TIR-derived peptides with differing inhibitory capabilities.

3) *Selection of P7 for further studies was made based on its ability to co-IP with TRAM (Fig 1). Is P7 optimal of the series for inhibition of MyD88-mediated TLR signaling?*

Yes, P7 was selected based on its ability to block SLAMF1-TRAM co-immunoprecipitation as it was initially designed to target these protein-protein interaction. However, as could be seen from the Supplementary figure 1B, which shows the results of the screening of different SLAMF1 peptide variants in THP-1 macrophages-like model system, P7-Pen was the most efficient peptide to block both LPS-mediated TLR4-TRAM-TRIF signaling axis (CXCL10 secretion as a readout) and as well TLR4-TIRAP-MyD88 dependent proinflammatory cytokines secretion (TNF, IL-6, IL-1β). Thus, the P7 peptide is also optimal for inhibition of MyD88-mediated signaling, and we have emphasized this point in the manuscript (inserted text was highlighted).

4) *Does P7-Pen inhibit the ODN1668-stimulated TLR9? ODN1668 induces proinflammatory cytokines but reportedly does not induce much of IFN response.*

It is a very relevant and interesting question, which we have not addressed in the current study. Here we decided to check ODN2006, which induces both IFNβ and TNF response in our model system. Due to the suggestion of the Reviewer 2 to exclude these data from the manuscript, we decided to postpone these experiments for the following studies, where we would address the regulation of TLR9 signaling along with signaling via other endosomal TLRs by P7 peptide more precisely, trying different model systems and ligands.

5) *Labeling of blots shown in Fig 4 is confusing. Why do peptide treatments so strongly augment TRAM expression in WCLs?*

Labeling of blots on Figure 4 was corrected to include all background information about the transfection conditions, proteins' molecular weight and peptides' treatment. Whole cell lysates (WCL) represent input for the IPs, with additional information on input-IP ratio included to the figure legends as suggested by the Reviewer 2.

There was no difference in TRAM levels between the lysates (WCLs) from cells treated by C3 control peptide or P7 on Figures 4A, 4B and 4D. Some difference in Figure 4C could represent slightly unequal loading of the lysate/input sample.

6) "...opsinized particles for 45 min, flowed by..."

Abovementioned text was corrected in the legend for the Figure S3.

7) What are the "matching water treatments" for data in water-treated columns in Table S2?

There were two major experimental conditions in the described animal experiment: 1) animals that got peptide before LPS (pre-treatment groups) and 2) animals that got peptide in 30 min after LPS (post-treatment groups). Both groups included control water-treated animals (solvent control). We have modified the text in the Table S2 legend to clarify this.

Answers to Reviewer #2 comments:

1. *In the introduction, please rephrase the statement about the binding of PAMPs (or DAMPs) to the extracellular domain of PRRs, considering that some TLRs may occur both at the cell membrane and intracellularly.*

The domain title was corrected in the text of introduction to “N-terminal ligand recognition” domain (highlighted in the text).

2. *For the methods related to Fig.1a and Fig.6b, please indicate the polyacrylamide gel concentration used.*

The following missing information was included to the Western Blotting paragraph in Materials and methods section:

For SDS-PAGE, we used pre-cast gradient 4–12% Bis-Tris protein gels NuPAGE™ Novex™ and either MOPS (for resolution of 25-200 kDa proteins) or MES (for resolution of 2-100 kDa peptides/proteins) SDS running buffer (20X) (ThermoFisher Scientific, Norway).

Thus, for both figures we used above-mentioned gels, but for Figure 1a MOPS running buffer (resolution of proteins 20-80 kDa), and for Figure 6B, and Figure 8 we used MES buffer (to resolve very small biotinylated peptides).

3. *In Fig.1a, there is a doublet in the Western blot panel of Flag/TRAM(Flag). Please indicate the band corresponding to overexpressed TRAM.*

There is a weak upper band in lysates and IPs in Figure 1a, and we consider that both bands represent TRAM^{FLAG} protein. Depending from the backbone of the plasmid encoding for TRAM – TRAM with YFP tag (from K. Fitzgerald, University of Massachusetts Medical School, Worcester, MA) as used in Figure 1B or pCMV-(DYKDDDDK)-N (Takara) as in Figure 1A, TRAM was always expressed as single or double bands, respectively. This could happen probably due to some differences in the linker region between C-terminal Flag tag and TRAM that results in additional posttranslational modifications of the tagged protein in the used Flag vector.

4. For the lower panels (co-IP results) in Fig.1a, it would be helpful if the authors provide a figure with all samples in the same gel.

Figure 1A represents the cropped parts from the same gel and membrane, which is now indicated in the Figure 1 legend. The original image for the whole membranes is shown in Source data for Figure 1 and available for evaluation.

5. In Fig.1b 4a and 4b, the images should be adjusted adequately to preserve an adequate amount of background information.

We have tried to adjust the images as suggested by the reviewer, and in addition provided the original source files for both inputs (WCLs) and the bands in IPs in the respective Source files (Source file for Figure 1, Source file for Figure 4). Also, we have adjusted the labeling for the transfection conditions for all figures showing precipitations from transfected HEK 293T cells (Figure 1b, Figure 4, Figure 7D and 7E).

6. For Fig.1C, it would be beneficial to discuss why the authors did not perform a similar analysis for P7-Pen.

Thank you for the comment, the following sentence was added to the paper text (highlighted in the text of the manuscript):

Inhibition of LPS-mediated STAT1 phosphorylation by P7 peptide in THP-1 cells directly correlated with IFN β mRNA expression (Fig. 1C-D), thus, we further proceeded with qPCR analysis for evaluation of peptides' efficacy.

7. Provide more details regarding the methodology used for the development of the P7 peptide. Include information about how the single amino acid substitutions or additions were theoretically planned and defined, as well as the synthesis, purification, and characterization of the peptide. This will enhance the reproducibility of the study.

Information on peptides synthesis and purity (peptides were ordered, not synthesized in house) is presented in the Materials and methods section. Data for several additional peptides was included to the Figure 1B and sequences to the Table S1. All other details were described in the text of the manuscript (highlighted in the manuscript):

Page 7:

The synthesis of peptides was commissioned from specialized providers, adhering to stringent criteria for high purity (> 90-95%), and necessitating the conversion of the toxic Trifluoroacetic acid (TFA) salt to acetate salt after synthesis. This conversion is essential for the utilization of the peptide with living cells.

Page 8:

To pinpoint the amino acids that are essential for the interaction between the peptide and target proteins, we performed tests utilizing peptides where each individual amino acid in the P7 sequence was replaced with alanine - a small, nonpolar amino acid with a short side group. Additionally, we explored various substitutions involving nonpolar amino acids, specifically valine and alanine (at positions 3 and 5, respectively), by replacing them with leucine – a nonpolar amino acid characterized by a larger side group.

Page 8:

Most of the tested alanine substitutions led to a reduction in the inhibitory activity of the SLAMF1-derived peptide during the screening process (Fig S1B). Similarly, substitutions involving nonpolar amino acids with polar amino acids, such as serine and threonine, also resulted in a notable decline in the peptide's inhibitory effectiveness (Fig S1B). Meanwhile, positions 4 and 10 within the P7 peptide exhibit greater flexibility and could potentially be utilized to introduce diverse chemical modifications, aiming to enhance the peptide's stability in biological fluids.

8. Consider removing the data for the effects of the peptide on TLR9 signaling, as it may be unnecessary.

Thank you for this suggestion. We have decided to remove the data on TLR9 signaling from this manuscript.

9. In Fig.3A and B, the authors should discuss why human primary monocytes pretreated with P7-Pen and further exposed to bacteria do not reconcile the data on CCL3 and IL-8 when treated with LPS.

We have added the following explanation to the text on Page 10:

Similarly, P7-Pen exhibited a significant reduction in *E. coli*-mediated secretion of IFN β and IL-1 β across all tested concentrations. However, its inhibitory efficacy was less pronounced in the case of *E. coli*-mediated TNF and IL-6 secretion, with significant reductions observed only at the highest tested concentrations of P7-Pen (Fig. 3B). Notably, there was minimal alteration in the levels of MIP-1 α /CCL3 and IL-8 secretion. *E. coli* bioparticles could trigger recognition by various PRRs found within immune cells in whole blood, including TLR4, TLR1/TLR2, TLR2/TLR6, intracellular nucleic acid sensors, and the complement system [Mogensen, 2009; Wang, 2010; Lappegård, 2009]. The relatively diminished effectiveness of P7-Pen in inhibiting *E. coli*-mediated cytokine secretion, as compared to LPS-mediated signaling, could be attributed to its selective inhibition of TLR4-mediated signaling while not affecting complement-, TLR2- or other PRR-mediated pathways.

10. The text on the beginning of page 10 can be improved for brevity. There is no need to reintroduce the knowledge already covered in the introduction section.

Some of the text was shortened in the results section on page 10 as suggested by the reviewer.

11. In the co-precipitation and pull-down experiments, it would be helpful if the authors provide a Western blot image of a percentage of the input, whenever possible.

All immunoprecipitations and PDs have the input on the WB, which is titled as WCLs (whole cell lysates). The sample from the same lysates as used for IPs or PDs are shown on the figures. We have included information about the input/WCLs percentage loaded to the gel in the respective Figure legends.

12. The first paragraph of the Discussion section seems unnecessary, as the authors have objectively discussed those points in the Introduction section. The text can start from the second paragraph.

Thank you very much for your suggestion, the text was partially reduced to remove the information that was already mentioned in the introduction. Several other points were additionally included to the Discussion as suggested by Reviewer 1.

September 13, 2023

RE: Life Science Alliance Manuscript #LSA-2023-02164-TR

Dr. Maria Yurchenko
Norwegian University of Science and Technology
Centre for Molecular Inflammation Research (CEMIR)
Olav Kyrres gate 10
NTNU Campus Øya, Kunnskapsenteret, 3rd Floor
Trondheim, Sør-Trøndelag 7491
Norway

Dear Dr. Yurchenko,

Thank you for submitting your revised manuscript entitled "Peptide derived from SLAMF1 prevents TLR4-mediated inflammation in vitro and in vivo". We would be happy to publish your paper in Life Science Alliance pending final revisions necessary to meet our formatting guidelines.

- please address Reviewer 1's remaining comments
- please add the Twitter handle of your host institute/organization as well as your own or/and one of the authors in our system
- please make sure the author order in your manuscript and our system match;
- please be sure to add all authors in the authors' contribution section
- please move your main, supplementary figure, and table legends after the references section in the main manuscript text
- please update the legend for Figure S4 to include all panels
- figure 9 has only panels A-D, and there is a call-out for Figure 9E - please correct accordingly
- please add a callout for Figure S4A to your main manuscript text
- please incorporate the Supplementary materials and methods into the main Materials and Methods section

A. FINAL FILES:

B. MANUSCRIPT ORGANIZATION AND FORMATTING:

Sincerely,

Reviewer #1 (Comments to the Authors (Required)):

The manuscript of Nilsen et al. presents a valuable study that contains interesting new data and adds to the field of CPP TLR inhibitors. However, I felt that the text would benefit from additional editing and, in some instances, contains contradictory statements. I also felt that questions 1, 2, and 4 were not fully addressed. Please find additional comments embedded in the point-by-point response below.

Answers to Reviewer #1 comments:

1) The statement that "LPS binding induces homodimerization, causing conformational changes in the intracellular TIR domains..." requires a reference to the primary data.

Thank you for this correction, the reference (Bovijn, 2012) was included to the text of the manuscript.

Unfortunately, I was unable to find a discussion or data referring to a conformational change in TIR domains in the cited article. Please indicate specific data of the paper that show that "conformational changes in the intracellular TIR domains" indeed occur as a result of LPS binding.

2) A 2020 review on cell-permeable peptide inhibitors of TLRs listed ~3 dozen of such inhibitors (and more were discovered since), including peptides that bind TIRAP and TRAM. Authors should discuss how P7 effects are different from the effects of previously reported peptides.

Thank you for this comment, it is indeed very important to emphasize what are unique and common features of P7 peptide when compared to other peptides developed to inhibit TLRs' signaling pathways.

The discussion part of the manuscript was edited to include these points (inserted text is highlighted in the manuscript):

Inhibition of TLR4-mediated signaling has received a lot of attention in recent years. Many TLR4-targeting drugs are under development and/or in clinical trials, including TLR4 antagonists and small molecules such as TAK-242 (Resatorvid) that binds to TLR4 and interferes with its interaction with two adaptor molecules, TIRAP and TRAM (52, 53).

The clinical trial of Resatorvid as a sepsis treatment was indeed attempted, but the study was terminated for inefficacy of the agent.

Promising pre-clinical studies have been conducted with TIR-domain-derived peptides linked to CPP (13, 16, 17, 54). What are then the distinctive attributes of P7 peptide? To begin with, P7 is not made from the TIR or TIR-like domain.

The request was to discuss the P7 effects, not P7. It is obvious that peptides have different sequences.

As a result, its capability for multispecific binding to various TIR-domain-containing molecules is unlikely.

This statement directly contradicts to Authors' own finding that "The mechanism of action of P7 peptide is based on interference with several intracellular PPIs, including TRAM-SLAMF1, TRAM-RabFIP2, and TIRAP-MyD88 interaction."

This is in contrast to a defining characteristic of TIR-derived peptides (13).

Per the reference cited, TIR-derived peptides are a diverse group that cannot be defined by multispecificity alone.

Several other peptides from microbial or viral origins, unrelated to the TIR domain, were recently described, however, the region they target within TIRAP or TRAM adaptor proteins (as reviewed in (13)) does not overlap with that of P7.

Authors' definition of regions targeted by P7 is very broad: "P7 directly interacts with the N-terminal part of TRAM and TIR domain of TIRAP." In this definition, "the region they (other peptides from microbial or viral origins) target within TIRAP or TRAM adaptor proteins" does overlap with that of P7, contrary to the above statement.

Unlike other already developed inhibitory peptides (13), P7 does disrupt interaction of TRAM with TLR4 or TRIF, but rather block TRAM recruitment to TLR4.

This statement is unclear and unspecific. Please check and clarify.

P7 targets the N-terminus of TRAM-TIR, a region that exhibits low homology to other TIR-domain-containing proteins (referred to as region 2 in (13)) and is required for interaction with FIP2 and SLAMF1 proteins and TRAM trafficking ((19, 38)). Similar to certain TIR-domain-containing peptides such as 9R34- N, MyD88-BB, 2R9, 4R9, 6R9, and 9R11 (as reviewed in (13)), the interaction between the P7 peptide and TIRAP occurs within the TIRAP-TIR domain (amino acids 87-160). Simultaneously, the P7 peptide disrupts TIRAP's interaction with MyD88, yet it does not affect TIRAP's interaction with TLR4 nor does it inhibit TLR2-mediated signaling. This unique combination of effects sets P7 apart from the majority of the aforementioned TIR-derived peptides with differing inhibitory capabilities.

3) Selection of P7 for further studies was made based on its ability to co-IP with TRAM (Fig 1). Is P7 optimal of the series for inhibition of MyD88-mediated TLR signaling?

Yes, P7 was selected based on its ability to block SLAMF1-TRAM co-immunoprecipitation as it was initially designed to target these protein-protein interaction. However, as could be seen from the Supplementary figure 1B, which shows the results of the screening of different SLAMF1 peptide variants in THP-1 macrophages-like model system, P7-Pen was the most efficient peptide to block both LPS-mediated TLR4-TRAM-TRIF signaling axis (CXCL10 secretion as a readout) and as well TLR4-TIRAP-MyD88 dependent proinflammatory cytokines secretion (TNF, IL-6, IL-1). Thus, the P7 peptide is also optimal for inhibition of MyD88-mediated signaling, and we have emphasized this point in the manuscript (inserted text was highlighted).

4) Does P7-Pen inhibit the ODN1668-stimulated TLR9? ODN1668 induces proinflammatory cytokines but reportedly does not induce much of IFN response.

It is a very relevant and interesting question, which we have not addressed in the current study. Here we decided to check

ODN2006, which induces both IFN and TNF response in our model system. Due to the suggestion of the Reviewer 2 to exclude these data from the manuscript, we decided to postpone these experiments for the following studies, where we would address the regulation of TLR9 signaling along with signaling via other endosomal TLRs by P7 peptide more precisely, trying different model systems and ligands.

5) Labeling of blots shown in Fig 4 is confusing. Why do peptide treatments so strongly augment TRAM expression in WCLs?

Labeling of blots on Figure 4 was corrected to include all background information about the transfection conditions, proteins' molecular weight and peptides' treatment. Whole cell lysates (WCL) represent input for the IPs, with additional information on input-IP ratio included to the figure legends as suggested by the Reviewer 2.

There was no difference in TRAM levels between the lysates (WCLs) from cells treated by C3 control peptide or P7 on Figures 4A, 4B and 4D. Some difference in Figure 4C could represent slightly unequal loading of the lysate/input sample.

6) "...opsinized particles for 45 min, flowed by..."

Abovementioned text was corrected in the legend for the Figure S3.

7) What are the "matching water treatments" for data in water-treated columns in Table S2?

There were two major experimental conditions in the described animal experiment: 1) animals that got peptide before LPS (pre-treatment groups) and 2) animals that got peptide in 30 min after LPS (post-treatment groups). Both groups included control water-treated animals (solvent control). We have modified the text in the Table S2 legend to clarify this.

Reviewer #2 (Comments to the Authors (Required)):

I have reviewed the revised manuscript, and I am pleased to confirm that the authors have adequately addressed all of the concerns and inquiries raised. The revisions made by the authors have significantly improved the clarity, comprehensibility, and overall quality of the manuscript. Therefore, I recommend accepting this manuscript for publication in the journal.

We want to thank the reviewers for the evaluation of the revised version of the manuscript “Peptide derived from SLAMF1 prevents TLR4-mediated inflammation *in vitro* and *in vivo*”. In this letter we provide point-by-point reply to all the remaining reviewers’ comments.

Answers to Reviewer #1 comments:

1) The statement that "LPS binding induces homodimerization, causing conformational changes in the intracellular TIR domains..." requires a reference to the primary data.

Unfortunately, I was unable to find a discussion or data referring to a conformational change in TIR domains in the cited article. Please indicate specific data of the paper that show that "conformational changes in the intracellular TIR domains" indeed occur as a result of LPS binding.

Thank you for the comment, we have tried to be more specific with statements in the text and corrected the reference to the newest publication regarding the conformational changes of TLR4. Here is the corrected text:

LPS binding to TLR4/MD-2 triggers dimerization of the ectodomain and structural changes in TLR4 which lead to TIR-TIR dimer formation. In this agonistic TLR4 conformation, the TIR-TIR dimers bind to the sorting adaptor protein TIRAP which recruits the signaling adaptor MyD88. The detailed mechanism of TLR4 activation is not fully resolved on the atomic level, but a recent simulation study suggests a dynamic and plastic behavior of TLR4, which depend on the lipid environment (lipid rafts), and formation of two possible types of functional TIR-TIR dimers (symmetric and asymmetric) (7).

2) The clinical trial of Resatorvid as a sepsis treatment was indeed attempted, but the study was terminated for inefficacy of the agent.

Indeed, the clinical trial for Resatorvid as a sepsis drug was terminated, but Resatorvid is currently evaluated as TLR4-blocking drug for other clinical applications, thus, we think it is important to mention as a small molecule blocking TLR4 signaling.

The request was to discuss the P7 effects, not P7. It is obvious that peptides have different sequences.
Text was shortened and corrected accordingly.

This statement directly contradicts to Authors' own finding that "The mechanism of action of P7 peptide is based on interference with several intracellular PPIs, including TRAM-SLAMF1, TRAM-RabFIP2, and TIRAP-MyD88 interaction."

Statement is removed from the discussion part.

This is in contrast to a defining characteristic of TIR-derived peptides (13). *Per the reference cited, TIR-derived peptides are a diverse group that cannot be defined by multispecificity alone.*

This part is removed from the discussion text.

Authors' definition of regions targeted by P7 is very broad: "P7 directly interacts with the N-terminal part of TRAM and TIR domain of TIRAP." In this definition, "the region they (other peptides from microbial or viral origins) target within TIRAP or TRAM adaptor proteins" does overlap with that of P7, contrary to the above statement.

We have provided more precise definition of the P7-targeted region in TRAM. Region in TIRAP was defined in the text (87-160 amino acids in human TIRAP), and it was not yet narrowed down to the shorter sequence.

*"Unlike other already developed inhibitory peptides (13), P7 does **not (was in the manuscript text but absent in the comment)** disrupt interaction of TRAM with TLR4 or TRIF, but rather block TRAM recruitment to TLR4." This statement is unclear and unspecific. Please check and clarify.*

We have clarified the statement in the manuscript text.

Old text is in grey:

What are then the distinctive attributes of P7 peptide? To begin with, P7 is not made from the TIR or TIR-like domain. As a result, its capability for multispecific binding to various TIR-domain-containing molecules is unlikely. This is in contrast to a defining characteristic of TIR-derived peptides (13). Several other peptides from microbial or viral origins, unrelated to the TIR domain, were recently described, however, the region they target within TIRAP or TRAM adaptor proteins (as reviewed in (13)) does not overlap with that of P7.

Unlike other already developed inhibitory peptides (13), P7 does disrupt interaction of TRAM with TLR4 or TRIF, but rather block TRAM recruitment to TLR4. P7 targets the N-terminus of TRAM-TIR, a region that exhibits low homology to other TIR-domain-containing proteins (referred to as region 2 in (13)) and is required for interaction with FIP2 and SLAMF1 proteins and TRAM trafficking (19, 38).

Here is the corrected text in the discussion part (also highlighted by yellow in the manuscript):

What are then the distinctive attributes of the P7 peptide? Unlike other already developed inhibitory peptides (13), P7 does not disrupt the interaction of TRAM with TLR4 or TRIF, but rather prevents the trafficking of TRAM to TLR4, which in turn depends on the interaction of TRAM with SLAMF1 and FIP2 {Skjesol, 2019;Yurchenko, 2018}. P7 targets the SLAMF1- and FIP2-interacting motif in N-terminus of TRAM-TIR (69-95 amino acids in human TRAM), a region that exhibits low homology to other TIR-domain-containing proteins (referred to as region 2 in (13)) and is required for interaction with FIP2 and SLAMF1 proteins (19, 38).

September 22, 2023

RE: Life Science Alliance Manuscript #LSA-2023-02164-TRR

Dr. Maria Yurchenko
Norwegian University of Science and Technology
Centre for Molecular Inflammation Research (CEMIR)
Olav Kyrres gate 10
NTNU Campus Øya, Kunnskapsenteret, 3rd Floor
Trondheim, Sør-Trøndelag 7491
Norway

Dear Dr. Yurchenko,

Thank you for submitting your Research Article entitled "Peptide derived from SLAMF1 prevents TLR4-mediated inflammation in vitro and in vivo". It is a pleasure to let you know that your manuscript is now accepted for publication in Life Science Alliance. Congratulations on this interesting work.

DISTRIBUTION OF MATERIALS:

Again, congratulations on a very nice paper. I hope you found the review process to be constructive and are pleased with how the manuscript was handled editorially. We look forward to future exciting submissions from your lab.

Sincerely,
